# Club cells employ regeneration mechanisms during lung tumorigenesis

Yuanyuan Chen[1,10], Reka Toth [1,2,9,10], Sara Chocarro [1,3,10], Dieter Weichenhan[2], Joschka Hey [2,3], Pavlo Lutsik [2], Stefan Sawall[4], Georgios T. Stathopoulos [5,6], Christoph Plass [2,6,7,8] & Rocio Sotillo [1,6,7,8] ✉

The high plasticity of lung epithelial cells, has for many years, confounded the correct identification of the cell-of-origin of lung adenocarcinoma (LUAD), one of the deadliest malignancies worldwide. Here, we employ lineage-tracing mouse models to investigate the cell of origin of *Eml4-Alk* LUAD, and show that Club and Alveolar type 2 (AT2) cells give rise to tumours. We focus on Club cell originated tumours and find that Club cells experience an epigenetic switch by which they lose their lineage fidelity and gain an AT2-like phenotype after oncogenic transformation. Single-cell transcriptomic analyses identified two trajectories of Club cell evolution which are similar to the ones used during lung regeneration, suggesting that lung epithelial cells leverage on their plasticity and intrinsic regeneration mechanisms to give rise to a tumour. Together, this study highlights the role of Club cells in LUAD initiation, identifies the mechanism of Club cell lineage infidelity, confirms the presence of these features in human tumours, and unveils key mechanisms conferring LUAD heterogeneity.

Neoplastic growth of non-small cell lung cancer (NSCLC) is initiated by genetic and epigenetic alterations, occurring by transforming the cell of origin into a (pre)neoplastic cell state. During evolution of the cancer genome, additional alterations are acquired, which often accelerate tumorigenesis. These alterations are translated into unique gene expression profiles that determine the malignant phenotype, including aggressiveness and response to therapy. The altered molecular landscapes seen in cancer, the mixture of signatures of oncogenic processes and the cell of origin provide an opportunity for thorough molecular characterization of malignancies and biomarker development[1]. Recent studies on central nervous system tumours[2] and leukaemia[3–5] already harnessed this

concept for subclassification of brain tumours or outcome prediction of leukaemia therapy.

The human lung epithelium is composed of diverse cell types according to their location. The upper airways include Club, Ciliated, Basal, Goblet, Neuroendocrine, and Tuft cells as well as the recently described Ionocytes[6,7]. The distal airways include Alveolar type 1 (AT1) and Alveolar type 2 (AT2) cells. Many of these epithelial cells exhibit de-differentiation potential under normal homeostasis or upon lung injury. In the upper airway, Basal cells generate differentiated cells during postnatal growth[8]. Club cells maintain the airway by self-proliferation and differentiation into Ciliated cells[9] or Basal cells, but they can also mobilize to regenerate the alveoli after damage[10,11]. In the

[1]Division of Molecular Thoracic Oncology, German Cancer Research Center (DKFZ), Im Neuenheimer Feld 280, 69120 Heidelberg, Germany. [2]Division of Cancer Epigenomics, German Cancer Research Center (DKFZ), Im Neuenheimer Feld 280, 69120 Heidelberg, Germany. [3]Ruprecht Karl University of Heidelberg, Heidelberg, Germany. [4]X-Ray Imaging and CT, German Cancer Research Center (DKFZ), Im Neuenheimer Feld 280, 69120 Heidelberg, Germany. [5]Comprehensive Pneumology Center (CPC) and Institute for Lung Biology and Disease (iLBD), Helmholtz Center Munich-German Research Center for Environmental Health (HMGU), Max-Lebsche-Platz 31, 81377 Munich, Bavaria, Germany. [6]German Center for Lung Research (DZL), Heidelberg, Germany. [7]Translational Lung Research Center Heidelberg (TRLC), Heidelberg, Germany. [8]German Consortium for Translational Cancer Research (DKTK), 69120 Heidelberg, Germany. [9]Present address: Bioinformatics Platform, Luxembourg Institute of Health, Strassen, Luxembourg. [10]These authors contributed equally: Yuanyuan Chen, Reka Toth, Sara Chocarro. ✉e-mail: r.sotillo@dkfz-heidelberg.de

distal airway, both AT1 and AT2 cells repair the injured alveoli by self-renewal and differentiation into each other[12–14]. Besides these well-defined epithelial cells, a rare population of double-positive CCSP/SPC bronchioalveolar stem cells (BASCs), located in the terminal and respiratory bronchioles, can maintain both airway and alveoli[15,16]. Recently, single-cell RNA sequencing (scRNA-seq) has contributed to the identification of a rare population of H2-K1[high] Club-like progenitors[11], the transitional stem cell state of Krt8ADI[17], AT2-cell-derived damage-associated transient progenitors (DATPs)[18], and pre-AT1 transitional cells (PATS)[19], all of which contribute to lung regeneration. It remains controversial whether trans-differentiation of cells can take place during the initial transformation of normal cells, before giving rise to tumours. In this context, further characterization of this initial transition would be critical for the correct identification of the cell of origin of LUAD.

The cellular origin of NSCLC has been investigated over the last years using genetically engineered mouse models (GEMMs) demonstrating the importance of both the cell of origin as well as the genetic mutation spectrum in shaping lung cancer phenotypes[20]. For instance, many studies suggest AT2 cells to be the predominant cell of origin of *Kras*-driven LUAD[21–24]. However, rather than employing a stochastic system in which all cells have an equal chance of being transformed, these tumours were induced by forcefully expressing *Kras* in one specific cell type.

It is still highly debatable whether trans-differentiation of Club cells into AT2 cells can take place under specific environmental conditions or whether other genetic factors might alter the tumour-initiating cells[25]. We and others have demonstrated that Club cells can also induce LUAD. Employing engineered reporters together with toxic chemicals found in tobacco smoke, we proved that tobacco-induced tumours originate from airway epithelial cells, which attain alveolar characteristics, and not exclusively from cells originating in the alveoli[26]. In addition, two recent papers showed that mutant *Kras* expressed in Club cells induces LUAD in mice[27,28]. Altogether, these studies suggest that Club cells give rise to oncogene and chemical-induced LUAD by attaining an AT2 phenotype; however, there is a general lack of knowledge of the developmental routes and pathways that Club cells use to become a tumour.

As opposed to LUAD with *Kras* mutations, the cellular origin of *ALK*-translocated LUAD has not yet been addressed. In the present study, we investigate tumour initiation and development of Club cells by employing a stochastic LUAD model. We induce the endogenous oncogenic *Eml4-Alk* rearrangement using an adenoviral system[29] that can infect any cell type within the lung epithelium. We combine the *Eml4-Alk* adenovirus with lineage-tracing mouse models, DNA methylome, and single-cell RNA transcriptome analysis and demonstrate that Club cells upon *Eml4-Alk* fusion trans-differentiate early during tumour development, gain the expression of alveolar markers, and yield heterogeneous tumour subgroups by using their intrinsic regeneration mechanisms.

## Results

### Tumour initiation and development induced by *Eml4-Alk* fusion in distinct lung cell types

To faithfully recapitulate *ALK*-translocated human LUAD in mice, we used a published CRISPR-Cas9 construct to induce the endogenous *Eml4-Alk* (EA) oncogenic rearrangement[29] (Fig. 1a). In this model, intratracheal instillation of an adenovirus (hereafter Ad-EA) encoding a single guide RNA (sgRNA) targeting *Eml4*, a sgRNA targeting *Alk* and the Cas9 protein sequence, gives rise to LUAD 8 weeks after instillation. At the early stages of tumorigenesis, *Eml4-Alk* induced early hyperplasia in the bronchi that partially lost the Club cell marker CCSP, and acquired alveolar markers such as SPC, similar to *Kras* mutant LUAD, described by the authors in ref. 28 (Fig. 1b, c). Moreover, among more advanced tumour stages (adenoma and adenocarcinoma), almost every tumour

was SPC positive (Supplementary Fig. 1a), suggesting a lineage switch from Club to AT2-like cells triggered by the oncogenic transformation. Compared to those mice that only received Ad5-CMV-Cre (hereafter Ad-Cre) as control, we also found an increase of double-positive CCSP/SPC cells in the bronchiolar lesions 8 weeks after instillation of Ad-EA (Supplementary Fig. 1b). We speculate that this increase of dual positive cells in the early lesions is a consequence of a lineage switch from Club to AT2-like cells triggered by the oncogenic transformation and not necessarily an increase in BASCs, since these cells are localized in the bronchioles and not the BADJ (Supplementary Fig. 1c).

The advantage of using adenovirus to induce the *Eml4-Alk* rearrangement is that any cell type within the lung epithelium has the chance of being infected. To define the cell tropism of the adenovirus, we used Ad-Cre to instil a mouse strain that switches membranous tdTomato to membranous EGFP fluorescence (*mT/mG*)[30] upon Cre-mediated recombination. We identified that virtually all lung epithelial cell types got infected with Ad-Cre (Fig. 1d, e and Supplementary Fig. 1d), being AT1 cells the largest cell population infected, followed by AT2, Club, and Ciliated cells. We rarely found infected BASCs possibly due to their low frequency compared to the other cell types (Supplementary Fig. 1e and Supplementary Table 1).

From these results, we hypothesize that the *Eml4-Alk* oncogenic rearrangement can initiate tumours in Club and AT2 cells, although late-stage tumours homogeneously express the AT2 cell marker, SPC.

### Club and AT2 cells give rise to *Eml4-Alk* lung adenocarcinomas

To evaluate the cell of origin of *Eml4-Alk* induced LUAD we crossed *mT/mG* mice with *Scgb1a1-CreERT*[9] for labelling Club cells (hereafter *Scgb1a1*), *Sftpc-CreERT2-rtTA*[31] for AT2 cells (hereafter *Sftpc*), *Hopx-CreER*[32] for AT1 cells (hereafter *Hopx*) and *Foxj1-CreER*[33] for Ciliated cells (hereafter *Foxj1*). Next, we assessed the specificity of each lineage-tracing mouse line, by performing immunohistochemical and immunofluorescent staining of the corresponding cell markers together with GFP after tamoxifen-induced (TAM) recombination (Fig. 2a and Supplementary Fig. 2a). In line with the previous reports[11], the *Scgb1a1* line labelled not only Club cells (61% labelled), but also AT2 (11%) and Ciliated cells (23%) (Fig. 2b). Similarly, the *Hopx* line labelled Club (35%) and Ciliated cells (19%) in addition to AT1 cells (13%) (Fig. 2a, b and Supplementary Table 2). However, unspecific labelling was not found in *Sftpc*, that labelled 45% of AT2 cells, and *Foxj1* line that labelled 28% of Ciliated cells.

Subsequently, we induced LUAD in the lineage-tracing mice by intratracheal instillation of Ad-EA four weeks after TAM injection. When tumours became discernible by μCT, we analysed the percentage of GFP[+] tumours in each line (Fig. 2c–e and Supplementary Table 3). We found 44% ± 5 of GFP[+] lesions in *Scgb1a1* mice and 12% ± 9 in *Sftpc* (Fig. 2d, e and Supplementary Fig. 2b). Surprisingly, almost one-fifth of the tumours in *Hopx* mice were also GFP[+] (19% ± 14), although AT1 cells were rarely reported to contribute to LUAD[34]. Due to labelling promiscuity of the *Hopx* line, we further tested the possibility that these GFP[+] tumours were arising from Club rather than AT1 cells. Therefore, we infected *mT/mG; Krt5-CreER* mice (hereafter *Krt5*)[35] with Ad-EA, since only AT1 cells in the distal lung, besides Basal cells in the trachea, are labelled in this model (Fig. 3a, b and Supplementary Fig. 3a). However, we failed to observe any GFP[+] tumour in *Krt5* mice (*0 out of 130 analysed tumours*), suggesting that neither Basal cells from upper airways nor AT1 cells contributed to *Eml4-Alk* LUAD initiation (Fig. 3c, Supplementary Fig. 3b and Supplementary Table 3). Therefore, the GFP[+] tumours arising from the *Hopx* line most probably originated from unspecifically labelled Club cells (Fig. 3d, e). In addition, all tumours developed in *Foxj1* mice were negative for GFP (*0 out of 113 analysed tumours*) (Fig. 2d, e, Supplementary Fig. 2b and Supplementary Table 3). Notably, the GFP[+] tumours from *Hopx*, *Scgb1a1* and *Sftpc* lines exhibited the same expression pattern: positive for SPC but negative for CCSP (Fig. 3e and Supplementary Fig. 3c).

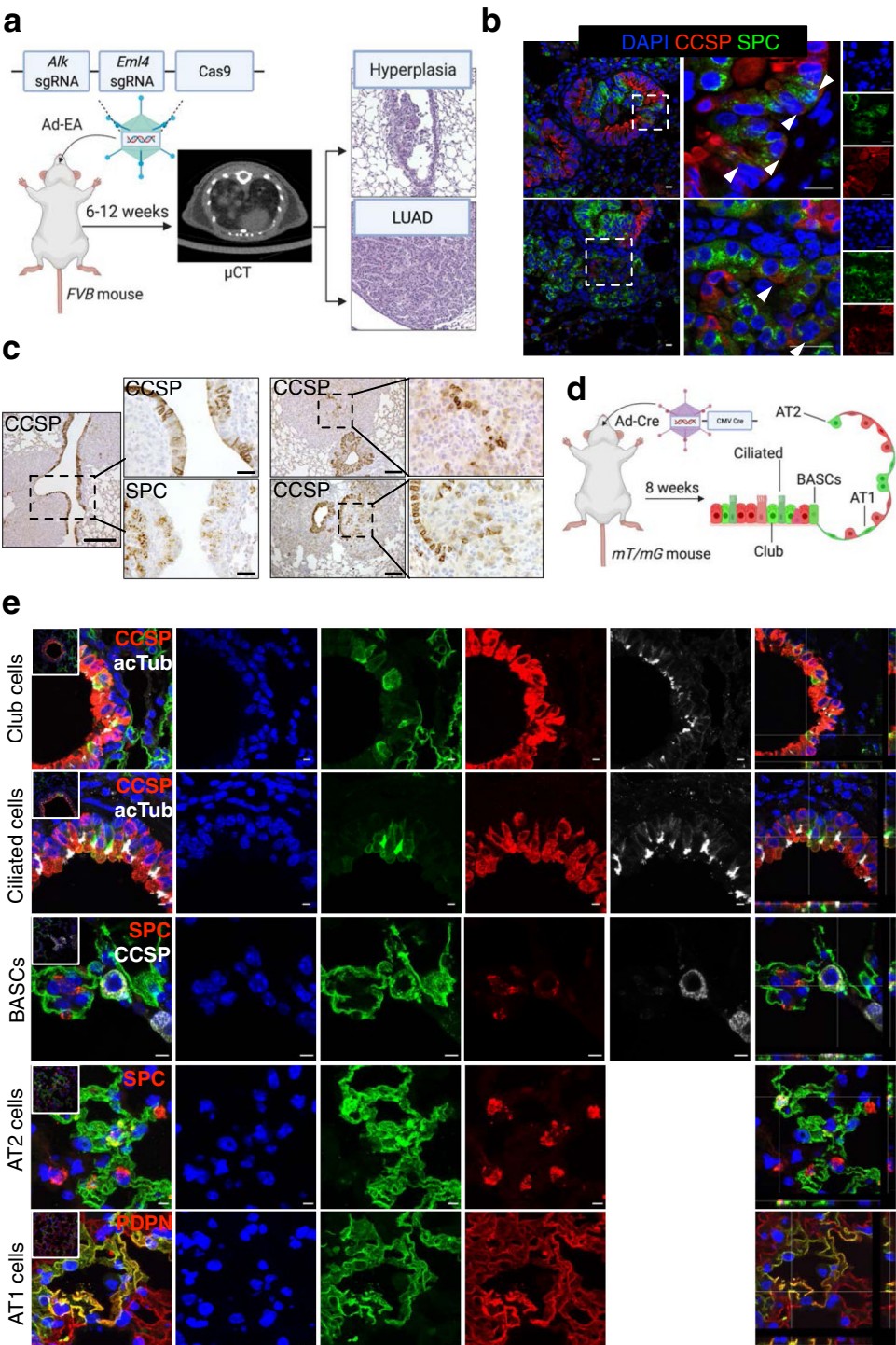

**Fig. 1 | Mouse model of lung adenocarcinoma and identification of adenovirus-infected cell types. a** Schematic of the *Eml4-Alk* mouse LUAD model used. Mice were intratracheally instilled with Ad-EA and µCT was used to monitor tumour development. Representative H&E staining of an early lesion (hyperplasia) and a tumour are shown on the right. **b** Immunofluorescent staining of SPC (green) and CCSP (red) antibodies showing that Club cells in the bronchi start expressing SPC upon *Eml4-Alk* rearrangement. DAPI staining in blue. Original (left), magnified overlay (middle), and their single-channel images (right) from the dashed areas are shown. Arrowheads show cells that are double-positive CCSP⁺SPC⁺. Scale bar:

20 µm. A minimum of 6 independent animals were analysed. **c** Immunohistochemistry of CCSP and SPC antibodies from *Eml4-Alk* tumours showing CCSP⁺ bronchi as well as cells inside the tumours. Scale bars: 200 µm (left panel) and 100 µm (right panel). A minimum of 6 independent animals were analysed. **d** Experimental schematic of *mT/mG* mice transduced with Ad-Cre indicating the cell types that can get infected. **e** Immunofluorescent staining of the indicated antibodies on lung sections from *mT/mG* mice transduced with Ad-Cre, showing that Club, Ciliated, BASCs, AT2 and AT1 cells are infected. Scale bars 10 µm. A minimum of 6 independent animals were analysed.

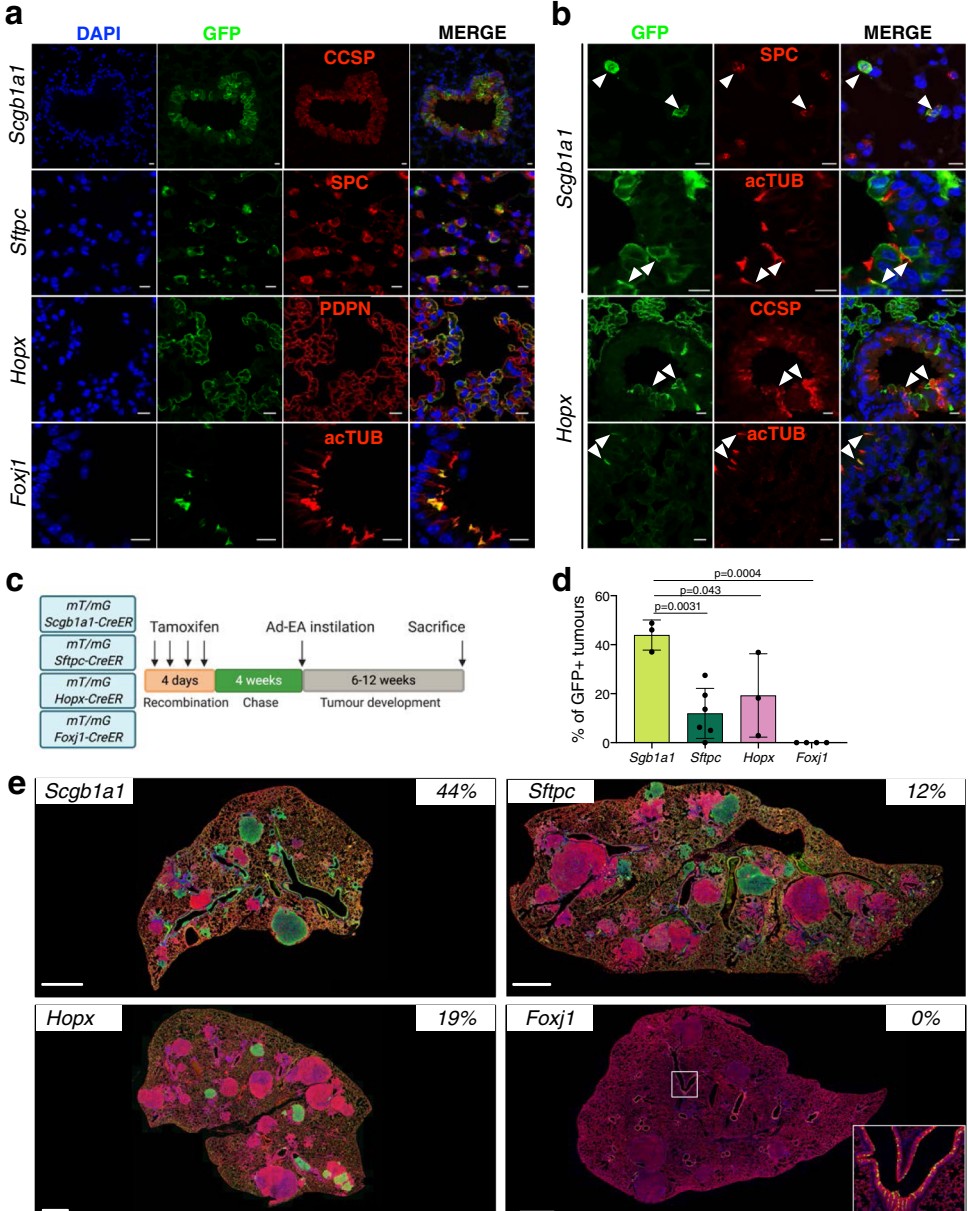

**Fig. 2 | Lineage-tracing models. a** Immunofluorescent staining of GFP and the indicated antibodies in the different lineage-tracing mouse models (*Scgb1a1*, *Sftpc*, *Hopx* and *Foxj1*) showing the specific labelling (CCSP, SPC, PDPN and acTUB). Images of single-channel and overlay are displayed sequentially. Scale bars: 10 μm. A minimum of 6 independent animals were analysed.
**b** Immunofluorescent staining of GFP and the indicated antibodies (CCSP, SPC and acTUB) in the lung sections from lineage-tracing mouse models showing the labelling specificity. Arrows indicate the unspecific labelling of the cells. Scale bars: 10 μm. A minimum of 6 independent animals were analysed. **c** Schematic of the labelling and tumour induction of lineage-tracing mice. **d** Percentage of

GFP⁺ tumours in the respective lineage tracing mice. *Scgb1a1*, 3 mice and 141 tumours analysed; *Sftpc*, 6 mice and 214 tumours analysed; *Hopx*, 3 mice and 117 tumours analysed and *Foxj1* 4 mice and 113 tumours analysed. One-way ANOVA, Tukey's multiple comparison test. Data are presented as mean values +/− SD. **e** Immunofluorescent staining of GFP and RFP antibodies in the respective mouse models. The percentages on the upper right corners represent the number of green labelled tumours out of the total number of tumours analysed in each line. Scale bars: 500 μm. Insert in the *Foxj1* example is provided to show the labelling of ciliated cells in the bronchi. A minimum of 6 independent animals were analysed.

Taken together, these findings establish both Club and AT2 cells as the cell of origin of *Eml4-Alk* induced LUAD. Additionally, the phenotypic similarity of the tumours suggests that Club cells might undergo a lineage switch upon oncogenic transformation.

**Lung cancer and cell-type-specific DNA methylation patterns**

Epigenetic reprogramming during differentiation is tightly regulated in a lineage-specific manner[5], resulting in cell-type-specific methylation[36]. Patterns of different developmental routes are preserved as "epigenetic memories" within each cell type[37]. Therefore, we

sought to uncover signatures of each specific cell type originated tumour and to trace them back to their cells of origin. We captured the DNA methylation landscape of sorted GFP⁺ cell populations from lineage-labelled healthy mice and tumours from the *Sftpc*, *Hopx* and *Scgb1a1* lines (Supplementary Fig. 4a) by tagmentation-based whole-genome bisulfite sequencing (TWGBS)[38].

Using TWGBS data, we determined regions which could distinguish different cell types based on DNA methylation differences. We found that poised enhancers, as opposed to active enhancers and promoters, were specifically powerful in discriminating normal lung

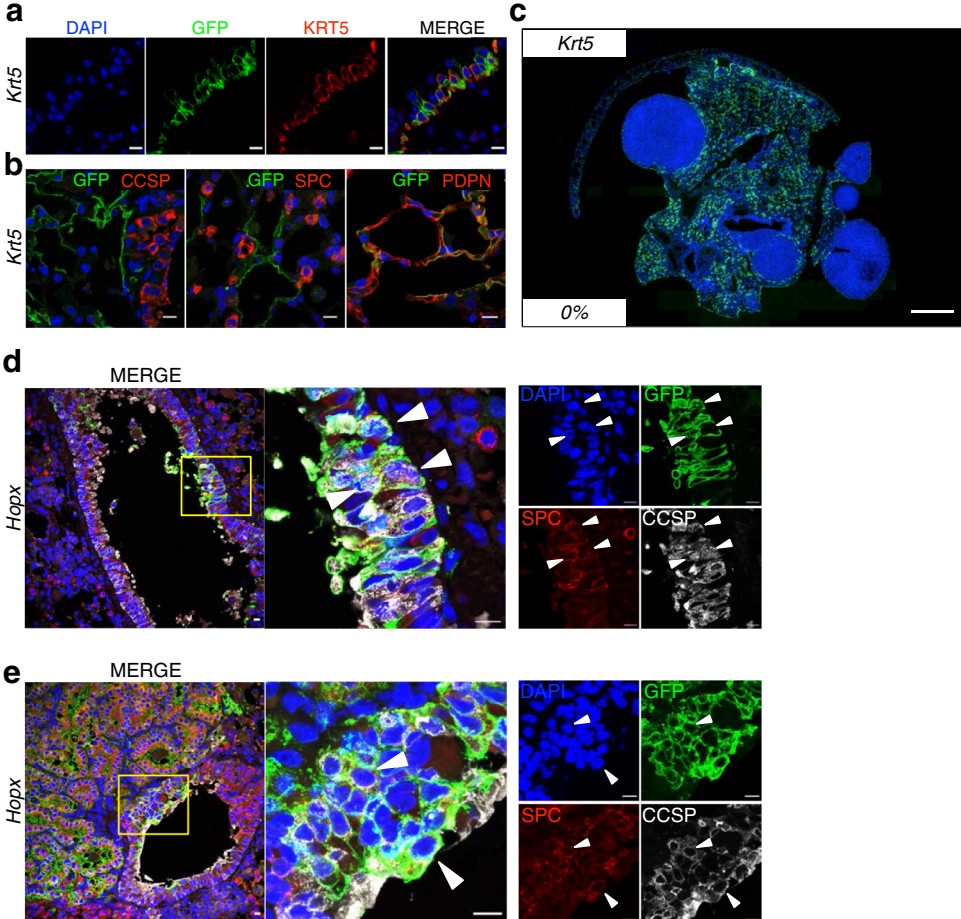

**Fig. 3 | Characterization of Krt5 and Hopx mice. a** Immunofluorescent staining of GFP and KRT5 antibodies in the trachea of *Krt5* mice showing labelled Basal cells. Overlay and single-channel images are sequentially presented. Scale bars: 10 μm. A minimum of 6 independent animals were analysed. **b** Immunofluorescent staining of GFP and the indicated antibodies in the distal lung of *Krt5* mice. AT1 cells, but not Club or AT2, are labelled. Blue (DAPI), green (GFP), red (as indicated). Scale bars: 10 μm. A minimum of 6 independent animals were analysed. **c** Overview of GFP immunofluorescent staining in *Krt5* mice. The percentage on the lower left corner represents the percentage of GFP⁺ tumours, 0 out of 130 tumours analysed in a total of 4 animals. Scale bars: 1 mm. A minimum of 6 independent animals were analysed. **d**, **e** GFP, SPC, and CCSP immunofluorescent staining on *Hopx* mice transduced with Ad-EA. Both early (**d**) and late (**e**) stages of tumorigenesis are shown; the original, magnified and single-channel images are sequentially shown from left to right; arrowheads indicate the labelled Club cells under lineage switch into AT2 cells. Scale bars: 10 μm. A minimum of 6 independent animals were analysed.

cell types. However, in principal component analysis (PCA), tumours showed homogenous methylation patterns and did not cluster according to their cellular origin, as did the normal cell types (Fig. 4a and Supplementary Fig. 4b). To infer cell type composition, we applied a reference-free deconvolution method, MeDeCom, to the DNA methylation data of the most variable CpG sites overlapping with bivalent enhancers. MeDeCom identifies so-called latent methylation components (LMCs) designed to capture the proportion of cell type-specific methylation signatures in each sample[39,40].

After optimization and quality control of the methylome deconvolution model (for details, see "Methods"), we identified three main LMCs: LMC1 and LMC3 represented normal samples from *Scgb1a1* and *Sftpc* lineages, respectively, whereas LMC2 represented a common lung cancer-specific signature (Fig. 4b). We could not identify a specific component for AT1 (appearing in *Hopx* lineage) or Ciliated cells (appearing in *Scgb1a1* lineage). To define the cell types represented by each LMC, we performed an enrichment analysis of the cell type-specific markers, previously identified by scRNA-seq[17,41,42]. Since promoter methylation is considered to correlate negatively with gene expression, the enriched gene signatures captured the cell types associated with the given component. We, therefore, correlated the DNA methylation levels of each promoter with the proportion of each LMC and determined whether cell type-specific markers were over-represented among the inversely correlating genes (Fig. 4c). The results revealed that LMC1 represented not only Club but also Ciliated cells. Similarly, LMC3 captured both AT2 and AT1 cells (Fig. 4c). Importantly, tumour samples, despite showing a homogeneous pattern in LMC2, still resembled the specific signature of their cell of origin, classified by a higher contribution (>10%) of the respective components (Fig. 4b). A high proportion (>10%) of LMC1 in both tumours from the *Hopx* line supported our previous result that these tumours are derived from Club cells.

Differential methylation analysis revealed cell-type-specific differences between AT2 and Club cells. Notably, tumours, irrespective of their cell of origin or the mouse line from which they originated, (*Scgb1a1*, *Sftpc* or *Hopx*), showed AT2-like methylation patterns (Fig. 4d). These results strengthened the assumption that Club cell originating tumours switch their identity during tumorigenesis. This was also supported by the hypermethylation of the *Scgb1a1* promoter in Club originated tumours, impairing the expression of this marker, and hypomethylation of the *Sftpc* promoter similar to that in normal AT2 cells, indicating the linage switch (Fig. 4e). To gain mechanistic insight into the lineage switch of Club cells during tumorigenesis, we analysed the enrichment of transcription factor (TF) binding motifs in differentially methylated regions (DMRs) (Fig. 4f and Supplementary Data 1). We observed strong enrichment of Forkhead family TFs among

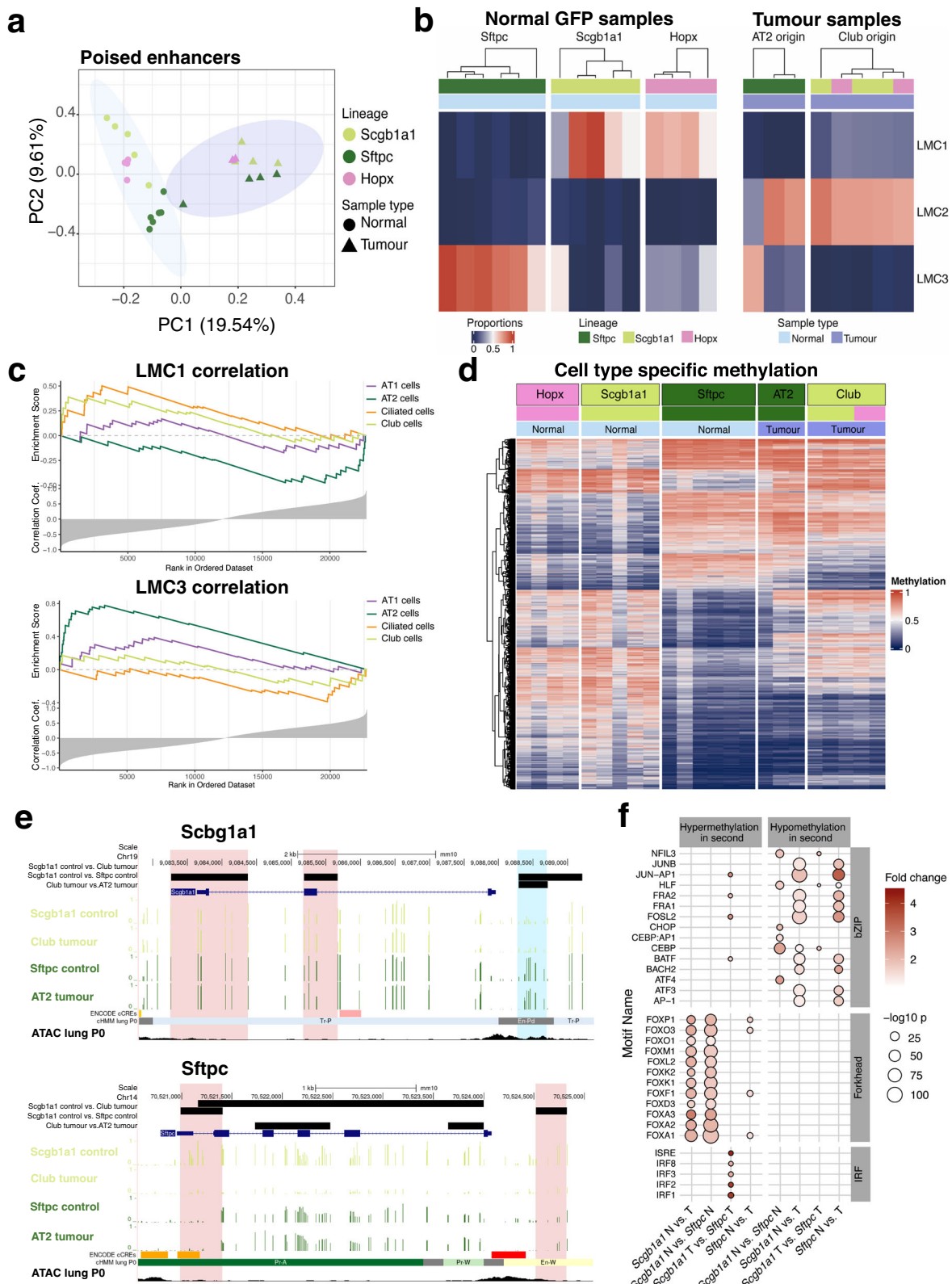

regions that were hypermethylated in Club cell originating tumours, as well as in normal AT2 cells compared to normal Club cells. Forkhead TFs regulate lung secretory epithelial fate[43] and play a role in maintaining epithelial cell identity[44]. Therefore, increased methylation of their binding site might contribute to derailing the normal lung regeneration processes. Tumour-specific hypomethylation events were enriched in TF binding motifs of the JUN/FOS proteins, building blocks of the activator protein-1 (AP-1) transcription factor known for

their role in lung tumorigenesis as well as their involvement within other signal transduction pathways[45].

Taken together, the DNA methylation data supports the Club cell origin of labelled tumours from *Scgb1a1* and *Hopx* lines. Furthermore, the DNA methylation patterns and differential methylation of TF binding motifs suggest an epigenetically driven lineage switch of Club cells into an AT2-like phenotype before or during tumorigenesis.

**Fig. 4 | DNA methylation patterns of normal and tumorigenic cells. a** Principal component analysis of DNA methylation at TSS-distal and TSS proximal poised enhancer regions (En-Pd and En-Pp, based on ENCODE postnatal 0 days mouse lung, ENCSR538YJF). The shape of the dots reflects the sample type, as normal lung (Normal) or GFP+ tumours (Tumour). The colour indicates mouse lineages used. Coloured shading is drawn around the Normal (light blue) and Tumour (purple) samples. **b** DNA methylation deconvolution by MeDeCom analysis shows three identified latent methylation components (LMC 1–3). Colours indicate the proportion of LMCs for each sample. Samples are clustered within each mouse lineage (*Sftpc*, *Scgb1a1* or *Hopx*) and according to sample type (Normal/Tumour). **c** Association of LMCs with cell type-specific gene signatures based on published single-cell RNA sequencing data[17]. Genes are ranked by their correlation of promoter methylation with the respective LMC. Enrichment scores are running sums calculated using the Gene Set Enrichment Analysis algorithm. **d** Heatmap of regions that were differentially methylated between *Scgb1a1* and *Sftpc* normal samples. The methylation value is shown as beta values ranging from 0 to 1 visualized by the heatmap colours blue to red. For tumours, the label of each column shows the proposed originating cell type, while the colours below depict their originating lineage. **e** Average methylation across the *Scgb1a1* and *Sftpc* genes for normal samples from *Scgb1a1* and *Sftpc* lines and Club and AT2 originated tumours. Pink bars highlight regions with DNA methylation differences between normal lineages, while blue bars highlight regions with differences in tumours from distinct origins. ENCODE cCRE, showing candidate Cis-Regulatory elements, cHMM lung P0 showing ChromHMM regions in mouse lung, postnatal 0 days, and ATAC lung P0 showing ATAC-seq peaks in postnatal 0-day-old lung are UCSC tracks under the same name. **f** TF motif enrichment analysis of the differentially methylated regions (DMRs). Each column represents one comparison (e.g. *Scgb1a1* normal vs. tumour) in one direction (e.g. hypermethylation in second, as in hypermethylated in tumours). The colour of the dots shows the enrichment of the DMRs for the motifs compared to random genomic regions. The size of the dots reflects on the $-\log_{10}(p$ value) of the enrichment analysis. Empty lines show the lack of significant enrichment.

## Single-cell RNA sequencing identifies multiple cell states after the transformation of Club cells

Lineage-tracing mouse models and DNA methylation data indicated that tumours originating from Club cells acquire an AT2-like phenotype. To elucidate the molecular processes of tumorigenesis, we designed a time-course experiment based on scRNA-seq of the *Scgb1a1* line (Fig. 5a). We collected two control samples; one with only TAM injection and another one 18 weeks after TAM and Adeno-Cas9 infection (Ad-Cas9); two early samples 2 weeks after Ad-EA and one intermediate sample 4 weeks after Ad-EA. We isolated live Club cells (EpCAM+/CD45−/CD31−/tdTomato−/GFP+/CD24−/ß4+/CD200+) from these samples (Fig. 5a and Supplementary Fig. 5a). To discern possible implications of the labelled AT2 cells in the *Scgb1a1* model, we included all GFP+ cells from a second intermediate sample—4 weeks after Ad-EA—as well as all GFP+ tumour cells from late-stage tumour nodules (Fig. 5a and Supplementary Fig. 5a) and performed scRNA-seq.

Unsupervised cell-clustering analysis of 29,277 high-quality cells uncovered 14 unique clusters (Fig. 5b and Supplementary Data 2), showing a distinct distribution of the samples throughout tumour progression (Fig. 5c, d). As indicated in Fig. 5d, clusters 1, 2, 3, 4, 5 and 8 were mainly (22–75%) composed of TAM and/or Cas9 control cells, while clusters 12, 13 and 14 were predominantly (>80%) composed of cells from the tumour sample. Clusters 6, 9, 10 and 11 were mainly composed (>55%) of cells derived from the 2 and 4 weeks transduced animals (Fig. 5c, d). Notably, the majority (74%) of cluster 7 came from the 4-week GFP sample, where labelled normal AT2 cells had not been excluded.

Next, clusters were annotated using cell-type-specific markers[42,46] and previously published cell type signatures (Supplementary Data 3)[11,17,47]. As expected, multiple clusters were annotated as Club cells and in line with our previous results, tumour clusters presented high levels of AT2 markers, except for cluster 13, which unexpectedly showed increased levels of AT1 markers (Fig. 5e, f and Supplementary Data 3). Furthermore, due to the high expression of *AW112010*, *H2-K1* and *Cd74*, we identified cluster 9 to be similar to the recently described H2-K1^high Club-like progenitor cells that were shown to contribute to lung regeneration after injury[11,18,19] (Supplementary Fig. 5b and Supplementary Data 3).

To verify whether tumour cells contained the *Eml4-Alk* rearrangement, we checked for the aberrant expression of the *Alk* region affected by the translocation. We found *Alk* expression not only in tumour clusters 12, 13 and 14 but also in cluster 11 (Fig. 5g). Moreover, we observed a progressive loss of Club cell identity and gain of AT2-like features throughout tumour development (Supplementary Fig. 6a). Notably the tumour sample contained very few cells still expressing the Club cell marker *Scgb1a1* (Supplementary Fig. 6b). Cluster 11 still showed a Club signature and low expression of tumour markers (Supplementary Fig. 6a, c, d), suggesting its pre-tumour stage.

Interestingly, this cluster showed expression of the Basal markers *Trp63* and *Krt5* (Supplementary Fig. 6e).

Collectively, our scRNA-seq analyses identified distinct cell populations that were enriched in different stages of Club cell tumorigenesis. A dynamic change in cell-type composition was observed with the rise of progenitor-like cell states in early time points and distinct tumour clusters being discernible as early as 4 weeks after oncogenic induction (Fig. 5h and Supplementary Table 4).

## Club cells employ lung regeneration mechanisms during tumorigenesis

We next sought to model the differentiation trajectory of Club cells towards LUAD. RNA velocity[48] estimates the ratio of spliced and unspliced mRNA predicting the future state of cells. Partition-based graph abstraction (PAGA) analysis[49] generates a map, where nodes are connected by weighted edges representing the connectivity between clusters. Using RNA velocity (Supplementary Fig. 7a) to direct the PAGA edges (Supplementary Fig. 7b), we obtained an unbiased estimation of linage trajectories[49,50] (Fig. 6a).

To better understand the transitions of Club cell states during tumour progression, we explored gene expression activity programmes using consensus non-negative matrix factorization (cNMF)[47,51]. We identified nine activity programmes (Fig. 6b and Supplementary Data 4). The intermediate Club-like progenitors (cluster 9) showed high activity of two immune-related programmes: an interferon programme and an immune activation/inflammation programme (Fig. 6c and Supplementary Data 4). Additionally, three programmes depicted the tumour clusters (12, 13 and 14) described above: a stem-like tumour module correlated with cluster 14, and was characterized by stem cell markers such as *Id2* and *Sox9* together with AT2 identity genes like *Sftpc*, *Lamp3* and *Lcn2* (Fig. 6b, c and Supplementary Data 4); a regeneration-like tumour module that was mainly enriched in cluster 12 and similar to lung regeneration, this module showed high activity in HIF1 signalling, TGF-beta pathway, IL-17 signalling and metabolic pathways such as glycolysis (Supplementary Fig. 7c and Supplementary Data 4)[18]. The third one, an AT1-like tumour programme, was highly active in cluster 13, although also present in cluster 12 (Fig. 6c). This module showed high similarity to recently described signatures implicated in lung regeneration[17–19] as well as to "mixed AT1/AT2" and "highly mixed" programmes described in AT2 originated *Kras* mutant lung tumours[47] (Supplementary Fig. 7d, e and Supplementary Data 3). These results suggest that *Eml4-Alk* Club cell-originated tumours are highly heterogeneous, similar to the *Kras* AT2 originated ones[47]. Since cluster 6 consisted of two separated cell types, we split the cluster into two for further trajectory analysis: 6 for the Activated Club cells and 6b for the Activated AT2s.

RNA velocity (Fig. 6a and Supplementary Fig. 7a) and PAGA analysis (Supplementary Fig. 7b) predicted cluster 9 to be

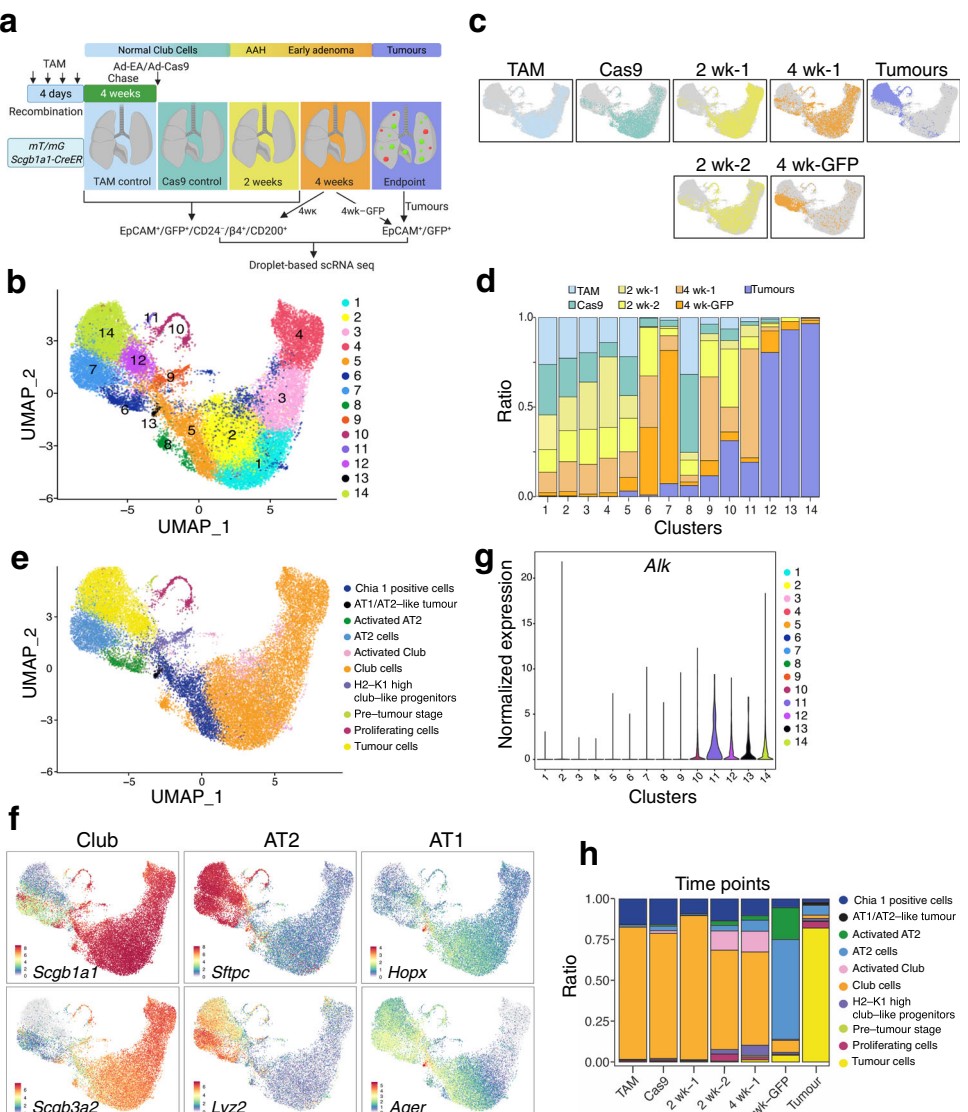

**Fig. 5 | Single-cell RNA sequencing. a** Experimental strategy of scRNA sequencing analysis. **b** UMAP of the high-quality cells. Cells are coloured by the clusters identified using a shared nearest neighbour (SNN) modularity optimization-based clustering. **c** UMAP embedding showing the distribution of the cells collected from the different time points. Coloured dots represent the cells collected at the indicated time point; grey dots represent cells from the other time points. **d** Percentage of cells in each cluster, coloured by sample origin. The relative contributions were normalized to the number of cells in each sample. **e** UMAP coloured by the cell-type annotation of the study set. The annotation combines previously published cell-type-specific markers[42,46] and manual curation and is based on the most significant marker genes of each cluster or cell group (Supplementary Table 5). **f** UMAP with cells coloured by expression of Club (*Scgb1a1*, *Scgb3a2*), AT2 (*Sftpc*, *Lyz2*) and AT1 (*Hopx*, *Ager*) cell-type-specific markers. Colours represent the normalized expression levels for each marker in each cell. **g** Violin plot depicting the normalized expression of *Alk* by cluster indicating elevated expression in clusters 11, 12, 13 and 14. **h** Cell-type composition of each collected sample.

connected to the tumour clusters. Therefore, we hypothesize that these Club-like progenitor cells could represent a crucial state in determining the fate of Club cell-originated tumours. Based on these results, we postulated that progenitor-like cells (cluster 9) with active interferon and inflammation signalling can follow two major routes towards tumorigenesis. The first one is through proliferating cells (cluster 10), before giving rise to tumour cells (cluster 14). The second route underwent the pre-tumour state (cluster 11) and further differentiated into tumour cells in clusters 12 and 13, with the latter also connected with cluster 9. Notably, the second route was also partly shared by a subpopulation of Club cells in cluster 4 that showed a Goblet-like gene signature (upregulated *Ltf*, *Bpifa1*, *Bpifb1*, *Reg3g* expression), and differentiated into cluster 11 (Supplementary Data 3), retaining the expression of *Ltf*. Our results show that although different tumour clusters arise on diverse paths, these paths are interconnected. As representatives of

these routes, we selected two candidate genes. *Ltf*, a key player of the second route, was highly expressed in clusters 4 and 11, although RNA velocity showed spliced transcripts in cluster 11 predicting its downregulation (Fig. 6d). The second candidate, *Trp63*, was highly expressed in cluster 11 (Fig. 6d and Supplementary Fig. 6e).

To further confirm these tumorigenic routes, we checked the expression of *Ltf* and *Trp63* in lung sections of different stages of tumour development and compared them to control mice instilled with Ad-Cas9 (Fig. 6e). In agreement with our computational analysis, we identified cytoplasmic *Ltf* expression in Club cells in early lesions, but not in normal Club cells or tumour cells, therefore validating the involvement of Club cells with Goblet-like (*Ltf*) signature in tumour progression (Fig. 6f). Additionally, few cells showed nuclear expression of *Trp63* in Club cells in early lesions, but not in normal Club cells or tumour cells (Supplementary Fig. 7f).

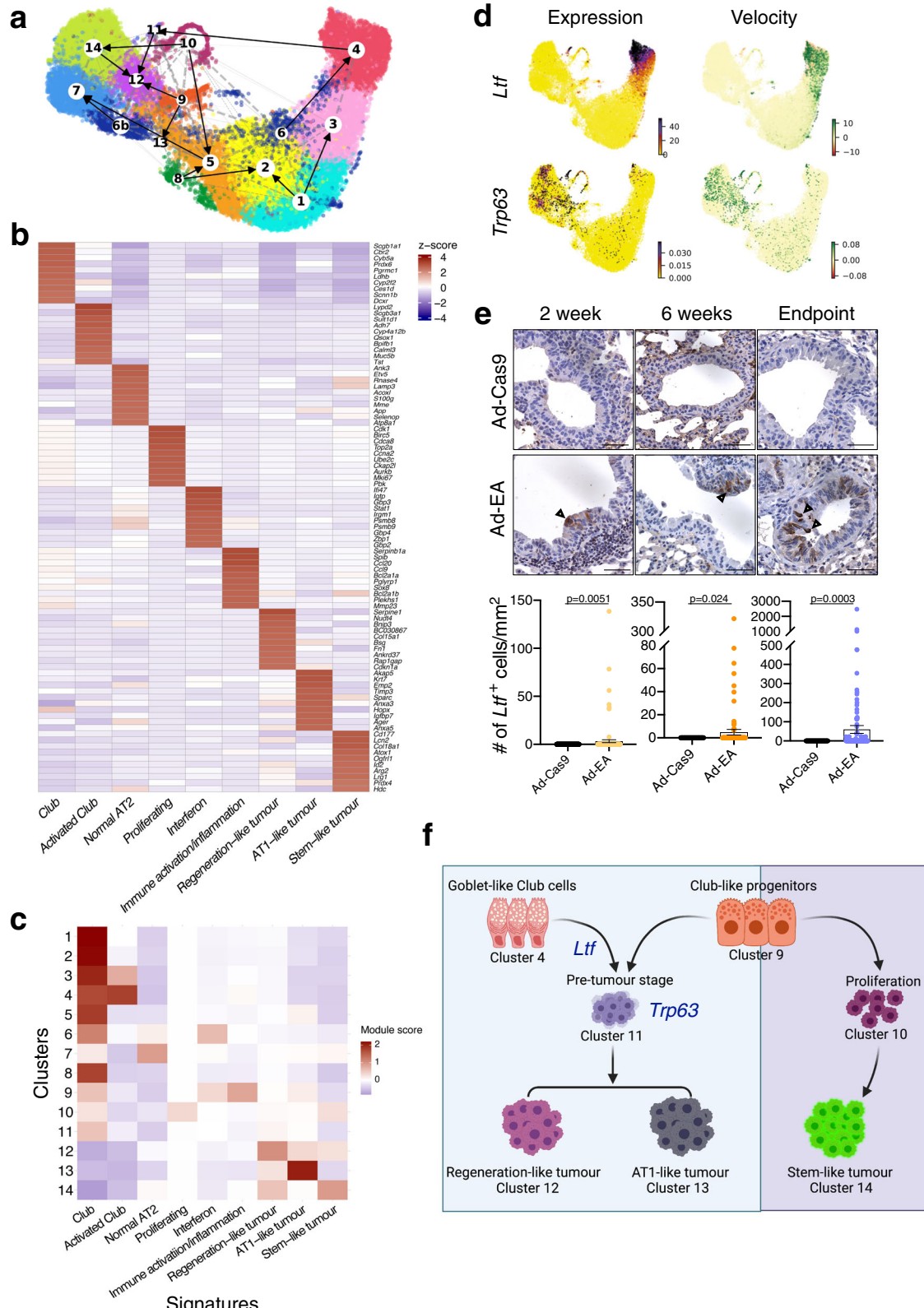

## Transcriptional activity programmes of Club originated tumours are present in human LUAD

Next, we analysed two published scRNA datasets on human LUAD. We compared tumour cells specific for 5 patients[52], and different transcriptional states of LUAD (Fig. 7a) and metastatic (malignant) cells[53] (Fig. 7b). We concluded that our tumour modules are present in LUAD and in non-lung metastasis, and show high variation between patients as well as between transcriptional states.

Then, we investigated the relevance of the 9 identified modules on LUAD patient prognosis taking advantage of the survival information of The Cancer Genome Atlas (TCGA) and bulk RNA-seq data[54]. We used the human orthologs of the module-specific genes to calculate the hazard ratio (Fig. 7c). We found that the Stem-like, AT1-like and Regeneration-like tumour signatures were associated with poor survival, with the last one being the most prominent indicator of poor survival.

**Fig. 6 | Activity programmes. a** UMAP embedding of combined RNA velocity and PAGA. Connections between clusters are established based on PAGA, while the direction of the arrows is inferred from the RNA velocity analysis. **b** Heatmap of the module scores of each cluster (*Y* axis) for the identified activity programmes (cNMF modules, "Methods"). The top 10 genes were used to calculate the module score. **c** Activity of the transcription programmes identified by the cNMF analysis for each identified cluster. The heatmap colours represent the *z*-score of the gene expression for the top 10 marker genes for each activity programme. **d** RNA velocity and gene expression of candidate genes *Ltf* (upper) and *Trp63* (lower). The left side of the plots shows the current gene expression levels. The right side of the plots shows the RNA velocity. The green colour represents a high velocity, therefore upregulation, while the red colour denotes downregulation. **e** Immunohistochemistry of *Ltf* in Ad-Cas9 control samples and in representative *Eml4-Alk* lung sections at different time points after adenoviral instillation. Scale bar 20 μm. Below, a quantification of the number of *Ltf* positive cells mm$^{-2}$ of bronchi of the animals. Each dot represents a bronchus. Mann–Whitney test, two-tailed. Data are presented as mean values +/− SEM. *n* = 3 mice per group were analysed except Ad-Cas9 2 weeks and Ad-EA 4 weeks (*n* = 4) and Ad-Cas9 6 endpoint (*n* = 2). **f** Schematic representation of the two routes followed by Club cells upon *Eml4-Alk* transformation.

Finally, to exclude that the observed signatures were simply tumour signatures and to validate whether Club originated tumours match better with LUAD than AT2 originated ones, we integrated our data with recently published 10× scRNA-seq data of AT2 originating *Kras* mutant;*p53* null tumours[55] (Fig. 7d and Supplementary Fig. 7g, h). We found marked differences between our main Club-cell originating tumour clusters and those of AT2 origin, while the AT1-like tumour cluster showed high similarity with the AT1-like cells. We also investigated how the gene expression module scores of our tumour clusters (AT1-like tumour, Regeneration-like tumour, and Stem-like tumour modules) are distributed among our clusters and the ones from ref. 55 (Fig. 7e). We found that both the AT1-like and the Regeneration-like tumour modules were higher in our clusters and were most similar to the AT1 and Late Gastric clusters from ref. 55 All tumour modules– including the Stem-like–scored significantly higher in our clusters. Altogether, these results suggest that the gene programmes associated with Club-originated tumorigenesis play a role in a subset of human LUAD.

## Discussion

The understanding of the cell of origin of lung adenocarcinoma is a debatable topic in cancer research. Here, we combined state-of-the-art lineage-tracing mouse models of lung cancer, DNA methylome and single-cell transcriptome analysis to unveil the tumorigenesis of Club-originated tumours. We showed that due to the high plasticity of lung epithelial cells, tumour characteristics are only partly dependent on their originating cell type. Previous studies focused on defining the cellular origin of LUAD, by using *Kras* mutant models that were based on tumour induction in specific cell types[21–23,26,28,56]. We studied the cell of origin of *Eml4-Alk* LUAD using a stochastic adenoviral system and employed lineage-tracing mouse models to label Club, AT2, AT1, Ciliated and Basal cells. We identified Club and AT2 cells as the main cell types implicated in *Eml4-Alk* LUAD development and showed that *Eml4-Alk* fusion gene in AT1, Ciliated or Basal cells did not give rise to tumours.

Remarkably, regardless of the originating cell type, all *Eml4-Alk* tumours were positive for the AT2 cell marker, SPC. In line with this observation, Rosigkeit et al.[28] have recently shown that Club-originated tumours become AT2-like by expressing SPC. DNA methylation was shown to act as cellular memory, storing information on previous differentiation states[37]. After deconvolution of the DNA methylomes from Club and AT2 originating tumours, cell type-dependent signatures were retained despite the large similarity between tumours. Although these signals could be due to contaminated normal cells in the tumours, we believe this is not the case, since based on the scRNA-seq analysis only very small number of cells in the tumour sample was expressing *Scgb1a1*, making it very unlikely that these cells generate a strong signal in the bulk methylome. In our study, analysis of the DNA methylome showed that Club tumours' methylation pattern was similar to that in AT2 cells, suggesting that Club cells switch their cellular identity and rewire their epigenomic landscape during tumorigenesis.

Our findings are especially interesting in comparison with cell-of-origin studies in haematological cancers. Normal haematopoiesis is characterized by unidirectional methylation changes[5,57] and leukaemia cells preserve large parts of these epigenomes[58]. This is in contrast to the patterns seen in LUAD where a dynamic, network-like differentiation landscape establishes highly similar tumour methylomes, arising on diverse pathways. Remarkably, the DNA methylation patterns in both cases can be used to infer the cell of origin.

Tumorigenesis can occur as a disruption of normal regeneration processes, as recently described in pancreatic cancer[59]. Club cells are able to regenerate the lung by renewing their own population[60] and transdifferentiating into AT2 and AT1 cells. Similar to lung regeneration, this high plasticity of Club cells was observed during *Eml4-Alk* tumorigenesis, where oncogenic stimuli rewired the transcriptome and epigenome of Club cells to evolve to an alternative fate programme. Our scRNA-seq identified several cellular stages which until now had only been shown to be transient states during lung regeneration after injury. For instance, an intermediate cell state derived from Goblet cells (expressing *Bpifb1*) dedifferentiated towards terminal Basal cells (*Krt5, Trp63*) upon lung regeneration[61]. Similarly, we observed that Goblet-like Club cells (cluster 4), marked by the expression of *Ltf* and *Bpifb1*, transition into a pre-tumour stage (cluster 11), which in addition to expressing high levels of *Alk*, shows the expression of *Ltf, Krt5* and *Trp63*. We speculate that our pre-tumour stage is similar to the Goblet-Basal intermediate stage found in lung regeneration and that Club cells mimic this state during tumour initiation. Additionally, we identified tumour-associated activity programmes marked by the expression of signatures also identified in intermediate states between Club or AT2 cells towards AT1 cells upon lung regeneration[17–19]. Our regeneration-like activity programme, associated mainly to tumour cluster 12, showed activation of HIF1 signalling, TGF-beta pathway, IL-17 signalling and glycolysis; pathways described to be essential in AT1 differentiation upon lung injury and regeneration[18]. Moreover, one of the oncogene-driven routes that Club cells follow involving cluster 9-11-12-13 shows high similarity to the signatures of Krt8ADI and PATS; two cellular states described in lung regeneration. Altogether, our data reinforce the importance of lung cell plasticity during tumorigenesis and stabilize a link between lung regeneration and lung tumorigenesis[17,19].

Finally, we found that all our identified tumour-related transcriptional programmes were present in human LUAD and associated with poor survival, the regeneration-like module being the most significant.

Interestingly, the tumorigenic routes followed by Club cells show correlation to some activity programmes previously reported in AT2-originated tumours[47,62,63]. This suggests that although lung tumours can have different cells of origin they can converge into similar stages of tumorigenesis.

By integrating scRNA-seq with whole-genome bisulfite sequencing analysis of *Eml4-Alk* tumours, we show that LUADs, unlike other tumour types, are only partly dependent on their originating cell type. In contrast, they are primarily determined by the oncogenic signals and further signalling pathways that are hijacked by the tumour cells. We identified that Club cells exploit signalling cascades that frequently govern normal lung development and regeneration, and we describe the paths that they follow resulting in a highly heterogenous state through two major tumorigenic routes. Characterization of these

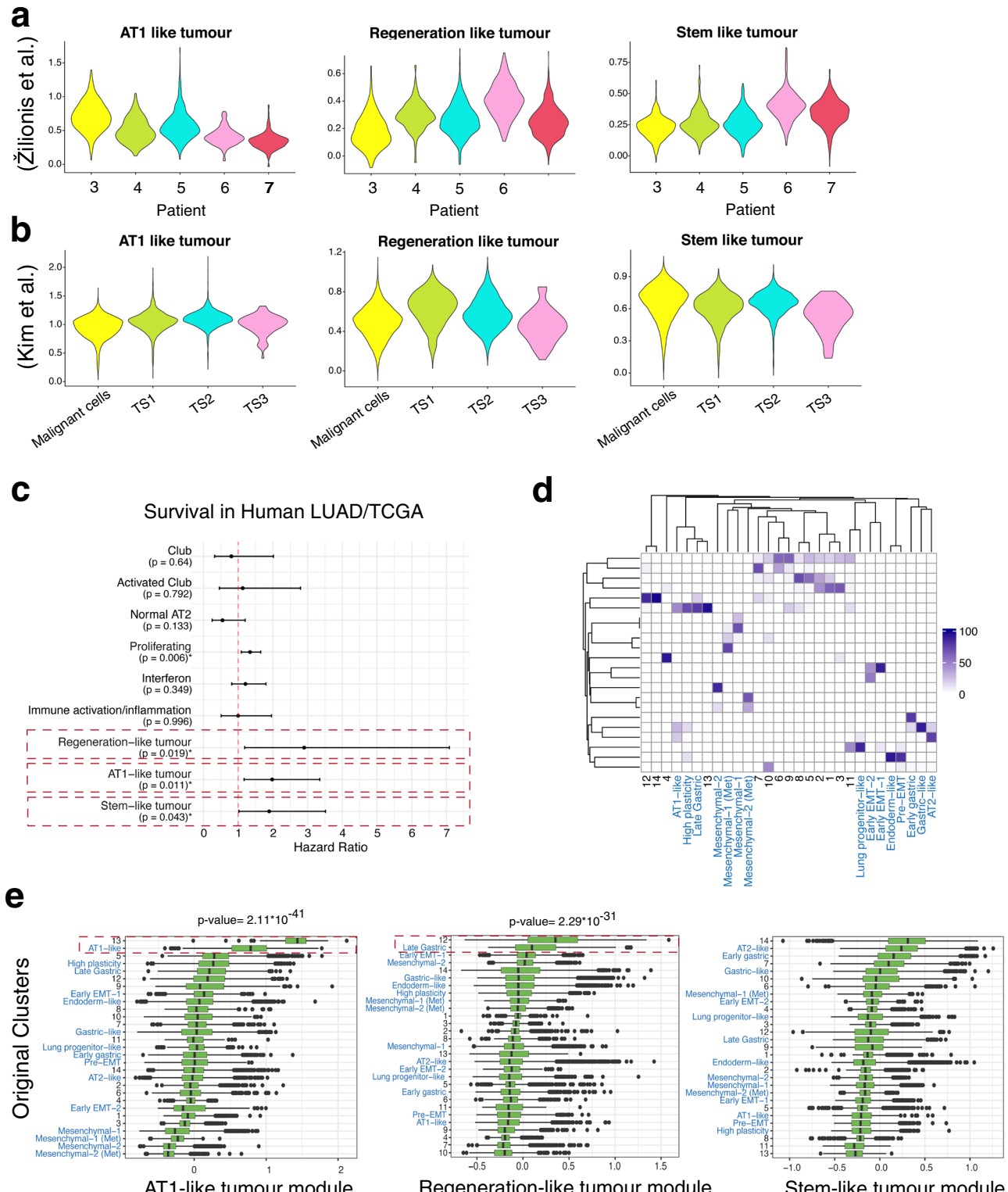

mechanisms, which are shared or unique in the different cancer subtypes, will help to decipher the routes of lung cancer initiation and identify actionable targets for personalized treatment of LUAD.

## Methods

### Mice, tamoxifen and adenoviral infection

*mT/mG* mice in C57BL/6 and FVB mixed background were kindly provided by Georgios Stathopoulos (University of Patras, Greece), *Hopx-CreERT2* line (ID: 017606) in 129 background and *Krt5-CreERT2* line in C57BL/6 and SJL mixed background (ID: 018394) were purchased from

The Jackson Laboratory. *Sftpc-CreER-rtTA* was kindly provided by Harold Chapman (University of California, San Francisco, US), and *Scgb1a1-CreERT*, *Foxj1-CreERT* in C57BL/6 background were kindly provided by Brigit Hogan (Duke University, Durham, US). All animals were kept in the above-mentioned background mixed with FVB. All mice were housed in specific pathogen-free conditions under a constant light–dark cycle and maintained on a standard mouse diet. All animal experiments had been approved by the local veterinary authorities and from the Regierungspräsidium Karlsruhe, Baden-Wurttemberg, Germany (animal license No. G185-17, G265-19). Requests for resources and reagents should be

**Fig. 7 | Validation of the tumorigenic routes. a** Violin plot depicting cNMF module scores in human LUAD based on single-cell transcriptome of patient-specific tumour cells. Patients 3 to 7 had a LUAD[52]. **b** Violin plot depicting cNMF module scores in human LUAD based on single-cell transcriptome. TS1-TS3 represent 3 tumour-specific transcription states of LUAD, and non-lung metastases labelled as "Malignant cells"[53]. **c** Forest plot depicting the Hazard Ratio (HR) and its confidence interval (CI) based on a Cox proportional hazards model in LUAD patients in TCGA for each gene expression module ($n = 533$). **d** Correlation-based heatmap of the cluster composition of our original clusters and the clusters from ref. 55. The rows are the integrated clusters, while columns are the original clusters from both studies. To avoid confusion, the clusters from ref. 55, are referred by name and coloured in blue. **e** AT1-like, Regeneration-like and Stem-like tumour module z-scores for cells belonging to the original clusters from this study and the one from ref. 55. The clusters from the Yang study are referred by name and coloured in blue. Cluster 13 vs. AT1-like, AT1-like tumour module, p-value $2.11 \times 10^{-41}$; Cluster 12 vs. Late Gastric, Regeneration-like tumour module $p$ value $= 2.29 \times 10^{-31}$. $n_1 = 3502$, $n_2 = 4710$, $n_3 = 3091$, $n_4 = 3027$, $n_5 = 2717$, $n_6 = 1706$, $n_7 = 2684$, $n_8 = 506$, $n_9 = 518$, $n_{10} = 594$, $n_{11} = 64$, $n_{12} = 1526$, $n_{13} = 103$, $n_{Mesenchymal-1} = 9950$, $n_{Mesenchymal-2} = 9028$, $n_{Mesenchymal-1 (Met)} = 818$, $n_{Mesenchymal-2 (Met)} = 3829$, $n_{Early EMT-1} = 3513$, $n_{Early EMT-2} = 460$, $n_{AT2-like} = 4680$, $n_{Pre-EMT} = 2108$, $n_{Early gastric} = 2175$, $n_{Lung progenitor-like} = 1074$, $n_{Gastric-like} = 4906$, $n_{Endoderm-like} = 10272$, $n_{Late Gastric} = 714$, $n_{High plasticity} = 2439$, $n_{AT1-like} = 1788$. The p values were calculated based on a two-sided $t$ test with the following results: $t = 20.677$, df $= 120.21$, 95% CI $= 0.554 - 0.671$ and $t = 5.2966$, df $= 1678.6$, 95% CI $= 0.0497 - 0.108$, respectively.

directed to and will be fulfilled by the lead contact, Rocio Sotillo (r.sotillo@dkfz-heidelberg.de).

To label epithelial cells in the lung, 6–12-week-old female and male mice were injected i.p. with $200\,\mu g\,g^{-1}$ tamoxifen (Sigma, T5648, $30\,mg\,ml^{-1}$ dissolved in corn oil) during 4 consecutive days. To select the optimal chasing period, we compared the number of labelled cells using 4 and 8 weeks chasing time in *Scgb1a1* mice. Since there were no differences in labelled cells in these two time points and previous data from the literature suggested that a period over 3 weeks was sufficient for TAM to label cells[64], we used 4 weeks of chasing time in all the experiments.

To induce *Eml4-Alk* rearrangement in the lung, mice were anaesthetized by intraperitoneal injection of $100\,\mu g\,g^{-1}$ ketamine and $14\,\mu g\,g^{-1}$ xylazine and intratracheally instilled with *Eml4-Alk* adenovirus, designed in the Ventura Lab[29] and purchased from Viraquest. The use of genetically modified organisms (GMO) was approved by the government of Baden-Wurttemberg, Germany (project No. 81078, 81155). Mice were randomly assigned to different experiments and investigators were not blind to the mice allocation during experiments and analysis. Mice were killed by cervical dislocation at the indicated time points or when the biggest tumour reached $0.5\,cm^3$. The maximal tumour size in the stated experiments was never exceeded. The lungs were perfused with 10 ml PBS through the right ventricle.

## Human samples
For the survival analysis, TCGA-LUAD RNASeq data was downloaded from the GDC Data Portal (https://portal.gdc.cancer.gov/, Data Release 24.0). For each cNMF module, the top 200 genes were selected. Using the human orthologs of these genes, we calculated a z-score on the primary tumours of TCGA-LUAD ($n = 533$). We used a Cox proportional hazard ratio model to examine the modules' effect on survival. The models were adjusted for age, gender and tumour stage. To visualize, we used a forest plot depicting the hazard ratio (HR) and the confidence interval (CI).

Human single-cell transcriptome datasets generated by Zilinois et al.[52] and Kim et al.[53] were downloaded to investigate the presence of cNMF modules in human lung cancer. Module scores were calculated as described by TCGA-LUAD.

## Immunostaining
Mouse lungs were incubated with 10% formalin (Sigma, HT501128) on a tube rocker for 24 h before further processing in a tissue processor (Leica ASP300S). Lungs were embedded in paraffin blocks and sectioned at 3 μm thickness. For immunofluorescent staining, the following primary antibodies were used: ProSP-C (Millipore, AB3786, 1:500), CC10 (Santa Cruz, SC-9772, 1:500), acetylated tubulin (Sigma, T7451, 1:500), Cytokeratin 5 (Abcam, ab53121, 1:200), Podoplanin (Abcam, ab11936, 1:200), GFP (Cell Signalling Technology, 2956S, 1:200), GFP (Abcam, ab5450, 1:200), RFP (Rockland 600-401-379, 1:200), Aquaporin 5 (Abcam, ab78486, 1:1000) and Scgb3a2 (1:500, R&D Systems, AF3465). Secondary antibodies were Alexa 488 donkey anti-rabbit IgG (Thermofisher, A21206, 1:500), Alexa 488 donkey anti-goat IgG (Abcam, ab150129, 1:500), Alexa 568 donkey anti-rabbit IgG (Thermofisher, A10042, 1:500), Alexa 568 donkey anti-goat IgG (Thermofisher, A11057, 1:500) and Alexa 568 donkey anti-mouse IgG (Abcam, ab175700, 1:500). Pictures were taken in a Leica SP5 confocal system and Tissuegnostic TissueFAX system. For immunohistochemical staining, the ABC kit (PK-6101) and DAB peroxidase substrate kit (SK-4100) from Vector Laboratories were used according to the manufacturer's instructions. Primary antibodies were: GFP (Cell Signalling Technology, 2956S, 1:200), CCSP (Millipore, 07-623, 1:1,000), SPC (Millipore, AB3786, 1:500), Lactotransferin (Sigma, 07-685, 1:500) and p63 (Sigma, ab735, 1:200). Pictures were taken in a Zeiss Axioplan microscope and Tissuegnostic TissueFAX system.

## Lung/tumour cell isolation and FACS sorting
For normal cells, lungs were minced into smaller pieces, and for tumour cells, visible nodules were carefully picked out, and the surrounding healthy tissue was removed. For TWGBS, both healthy lungs and tumours were dissociated into single cells using the lung dissociation kit (Miltenyi Biotech, 130-095-927) in a gentleMACS Octo Dissociator (Miltenyi Biotech, 130-095-937). After depletion of red blood cells with the red blood cell lysis buffer (Sigma, R7757), cells were incubated with a tumour cell isolation kit (Miltenyi Biotech, 130-110-187) following the manufacturer's instruction and passed through the LS columns (Miltenyi Biotech, 130-042-401). Cells from the flow-through were collected and DAPI ($1\,\mu g\,ml^{-1}$) was added as a viability marker. Cells were then sorted in a BD FACSAria cell sorter and the DAPI⁻;tdTomato⁻; GFP⁺ population was collected for downstream applications.

For single-cell RNA sequencing, lungs were perfused with PBS, injected with 1 ml digestion cocktail ($50\,U\,ml^{-1}$ dispase, $250\,U\,ml^{-1}$ collagenase, $5\,U\,ml^{-1}$ elastase, $30\,\mu g\,ml^{-1}$ DNAse I) through the trachea and cropped into small pieces. They were then incubated with 3 ml of digestion cocktail on a tube rocker for 30 min at room temperature before being dissociated into small pieces by plungers in 10 cm petri dishes. Samples were incubated 10 min with 5 ml DMEM with 10% FCS, 1% P/S and $100\,\mu g\,ml^{-1}$ DNAse I at room temperature. For endpoint tumours, visible nodules were picked out after adjacent healthy tissue was removed. Tumour cells were processed into single cells using the lung dissociation kit and red blood cells lysis buffer as described above. Tumour cells were then incubated with CD31 and CD45 microbeads (Miltenyi Biotech, 130-110-187) following the manufacturer's instructions and passed through the LS columns (Miltenyi Biotech, 130-042-401). Cells from the flow-through were incubated with DAPI ($1\,\mu g\,ml^{-1}$), CD45-PE ($1\,\mu g\,ml^{-1}$), CD31-PE ($1\,\mu g\,ml^{-1}$), EpCAM-BV711 ($1\,\mu g\,ml^{-1}$), CD24-BUV395 ($1\,\mu g\,ml^{-1}$), ß4-Alexa Fluor 647 ($5\,\mu g\,ml^{-1}$) and CD200-BV421 ($1\,\mu g\,ml^{-1}$) for further purification. Cells were then sorted in a BD FACSAria cell sorter and live EpCAM⁺; CD45⁻; CD31⁻; tdTomato⁻; GFP⁺; CD24⁻; ß4⁺; CD200⁺ population was collected for TAM, Cas9, 2wk-1, 2wk-2 and 4wk-1 groups and live EpCAM⁺; CD45⁻; CD31⁻; tdTomato⁻; GFP⁺ population was collected for 4wk-GFP, tumours. Collected cells were resuspended in 1000 cells μl⁻¹ before proceeding to library preparation.

## DNA isolation and library preparation for bisulfite sequencing

Lung cell dissociation and sorting of GFP+ cells were done as mentioned above. Each sample was a pool of 1–4 mice. DNA was isolated with QIAamp DNA Micro Kit (Qiagen, 56304). Tagmentation-based whole genome bisulfite sequencing (TWGBS) libraries were generated as described previously[38] using 20–30 ng genomic DNA as input. Per sample, four sequencing libraries with different barcodes were prepared and pooled in equimolar amounts to a final concentration of 2–10 nM. Pools were sequenced paired-end, 125 bp, on one lane of a HiSeq2000 sequencer (Illumina). Raw fastq files were aligned to the mm10 reference genome using methylCtools[65] as implemented by the Genomics and Proteomics Core Facility of the German Cancer Research Center.

## DNA methylation analysis

Methylation levels were called by MethylDackel (https://github.com/dpryan79/MethylDackel). Due to the specifics of the TWGBS, the following parameters were used to remove M-bias:−nOB 2,11,11,2−nOT 8,1,2,11. Quality control of the results included checking M-bias, bisulfite conversion rate and global methylation. Two samples were removed due to quality issues. Methylation data were analysed using R/Bioconductor 4.0 with packages Methrix[66] and bsseq[67]. CpG sites overlapping with single-nucleotide polymorphisms (SNPs) in any of the mouse strains were excluded based on data downloaded from Mouse Genome Project, database version 142[68]. The quality of the samples is shown in Supplementary Data 5.

For reference-free cell type deconvolution, we used MeDeCom, which allows the decomposition of DNA methylation into latent methylation components[39]. Regions were selected based on the 15-state ChromHMM model for mouse lung, postnatal 0 days downloaded from ENCODE[69,70] (ENCSR538YJF). The 100000 most variable CpG sites overlapping with poised or bivalent enhancers were included in the model. MeDeCom model was run using multiple lambda and K parameters with the following arguments: NINIT = 10, NFOLDS = 10, ITERMAX = 300. The final model with $K = 4$ and lambda=0.0001 was selected based on cross-validation error. Labelled tumours were re-categorized based on their suspected cell of origin. Tumours with >10% proportion in LMC1 or LMC3 were categorized as Club or AT2 originating, respectively. To identify gene promoters negatively associated with the LMCs, we calculated the Pearson correlation coefficient between LMC proportions and promoter methylation. Using this as ranking, we performed a Gene Set Enrichment Analysis (GSEA), as implemented in the fgsea package[71] with the following parameters: minSize = 3, maxSize = 500, nperm = 1000. For each set of marker genes, the running score was visualized.

Differential methylation calling was performed with the DSS package[72]. The data was smoothed with a smoothing span of 500. Dispersion of the groups was assumed non-equal. Regions were assigned as differentially methylated based on the following parameters: delta = 0, p.threshold = 0.001, minlen = 50, minCG = 3, dis.merge = 50, pct.sig = 0.5. DMRs were annotated using ChromHMM 15-state model for mouse lung, postnatal 0 days (ENCSR538YJF), using annotatr package[73]. Visualizations were created using ggplot2[74], ComplexHeatmap[75] and Gviz[76] packages.

Transcription factor binding motif enrichment analysis was performed with Homer[77] based on DMRs with the following parameters: -len 8,10,12 -size 100 -S 8 -cache 6921 -fdr 0. Enrichment analysis was performed as implemented in Homer. Multiple testing correction was done using false discovery rate (FDR) for all regions included.

## Single-cell RNA sequencing and data analysis

The time points and gating strategy used to collect the samples are shown in Fig. 5a and Supplementary Fig. 5A. Lung cells were dissociated as mentioned above and GFP+ cells were harvested as indicated in Supplementary Fig. 5A. Libraries were prepared using Chromium Next GEM Single Cell 3′ GEM, Library & Gel Bead Kit v3.1 (10× Genomics) according to manufacturer's instruction. Gene expression counts were acquired using Cell Ranger 3.1 count from 10× Genomics with its default settings. The gene expression counts were analysed in R 4.0.3 using Seurat 3.2.2[78] unless indicated otherwise. Based on visual inspection, the following number of detected gene thresholds were used to exclude bad quality cells among the samples: 1000–6000 for TAM, 1000–6000 for Cas9, 1000–6000 for 2 wk-1, 2000–6000 for 2 wk-2, 1500–6000 for 4 wk-1, 2000–6000 for 4-wk GFP+, 1000–6000 for tumour. Cells with more than 10% mitochondrial genes were removed from all samples. Samples were normalized using a scaling factor of 10,000. Variable features were selected using the FindVariableFeatures function of Seurat.

The cell-cycle stage of the cells was calculated using the CellCycleScoring function of Seurat with a publicly available dataset[79]. Genes were converted from human to mouse using the BioMart database with biomaRt R package[80,81].

Batch effect correction was done using Harmony[82] based on the PCA dimensionality reduction. Based on the resulted embeddings, we performed Uniform Manifold Approximation and Projection (UMAP)[83] and clustering. Clusters were calculated using a shared nearest neighbour (SNN) modularity optimization-based clustering algorithm. First, the $k$-nearest neighbours were calculated based on the first 30 dimensions of the Harmony embedding and an SNN graph was constructed. An optimal number of clusters was selected based on a suggestion by an elbow plot, as implemented in Seurat. Clustering was performed using the Louvain algorithm with a resolution of 0.6. Overexpressed markers of the clusters were selected if they were expressed in more than 25% of the cluster with a log fold change (FC) of 0.25.

After the initial analysis, a small cluster of Ciliated cells marked by high expression of *Foxj1* was identified and excluded. The above process was then repeated without Ciliated cells.

*Alk* expression was calculated by counting the reads overlapping with the part of *Alk* gene affected by the translocation (chr17: 71867045-71898183). The counts were lognormalized to the number of transcripts in each cell.

Consensus non-negative matrix factorization (cNMF)[51] v.1.2 was run using 100 iterations and the 2000 most variable genes. Based on the optimization, a model with 9 components was selected. For the consensus estimates of the programmes a local density threshold of 0.01 was used. The other parameters were used as default.

Overrepresentation analysis of the genes representing the modules based on KEGG[84] and WikiPathways[85] databases was performed using the clusterProfiler Bioconductor package[86].

Module and signature scores were calculated using the AddModuleScore function of Seurat based on the default parameters.

RNA velocity models the direction and speed of individual cells in the gene expression space by estimating the ratio of spliced and unspliced mRNA. Using this ratio, it predicts the future state of individual cells on a short timescale[48]. Here, we calculated the velocity using scVelo 0.2.3[50]. Since our data was batch corrected using Harmony the calculated velocity was projected on the UMAP based on the Harmony embedding.

Partition-based graph abstraction (PAGA)[49] generates a topology-reserving map of single cells. Here we used PAGA as implemented in scanpy 1.7.2[87]. For the standalone PAGA visualization, we used the algorithm on the Harmony embedding. We also used RNA velocity to direct the PAGA edges, as implemented in scVelo[49,50].

Data integration of the time-course data from ref. 55 was performed using reciprocal PCA analysis of the Seurat package based on the most variable 4000 genes. We used a resolution of 0.9 to find clusters in the integrated dataset. We used the AddModuleScore Seurat command to calculate the scores for each module. The modules consist of the top 200 genes for each cNMF module coming from our dataset.

## µCT imaging

µCT imaging was performed using the Inveon multi-modality µPET/SPECT/CT system (Siemens Medical Solutions, Knoxville, USA). Acquisitions covering the thorax and lungs were performed using a tube voltage of 80 kV and a tube current of 500 µA. A total of 360 projections were acquired over 360° with an integration time of 200 ms each. The detector was operated using a 4 × 4 binning mode resulting in a resolution of approximately 100 µm in the centre of rotation. Image reconstruction with isotropic resolution was performed using the Feldkamp algorithm with a Shepp–Logan kernel onto a 512 × 512 × 928 grid with appropriately sized voxels. Image analysis was performed using ImageJ.

## Statistics and reproducibility

The statistical information for the experiments is detailed in the text, figure legends, and figures. The correlation between LMCs and gene promoter methylation was calculated using Pearson correlation. If not otherwise stated, $p$ value < 0.05 was considered significant. False discovery rate (FDR) was used as a multiple test correction method, where appropriate. In this case, FDR $q$ < 0.05 was considered significant.

Boxplots if otherwise not indicated are showing with the lower and upper hinges the first and third quartiles (the 25th and 75th percentiles). The upper and lower whiskers extend to ±1.5 × inter-quartile range from the upper/lower hinge. Data points outside of these ranges are plotted individually as outliers.

Two samples were excluded from the DNA methylation analysis, based on quality issues. Mice were randomized to different time point groups without selection. The investigators were not blinded to allocation during experiments and outcome assessment. No statistical method was used to predetermine sample size.

## Reporting summary

Further information on research design is available in the Nature Research Reporting Summary linked to this article.

## Data availability

The single-cell RNA-seq and DNA methylation data have been deposited at the GEO database under accession code GSE176186 and are publicly available. TCGA-LUAD RNASeq data and clinical data on survival were downloaded from the GDC Data Portal [https://portal.gdc.cancer.gov/projects/TCGA-LUAD]. Single-cell RNA-seq published by ref. 55 was downloaded from Zenodo[88]. Publicly available human single-cell transcriptome datasets generated by Zilinois et al.[52] and Kim et al.[53] were downloaded from GEO [https://www.ncbi.nlm.nih.gov/geo/query/acc.cgi?acc=GSE127465] and [https://www.ncbi.nlm.nih.gov/geo/query/acc.cgi?acc=GSE131907], respectively. All other data are available within the Supplementary Data and Source data provided with this manuscript. Source data are provided with this paper.

## Code availability

All original code have been deposited at Github [https://github.com/tkik/Lung_CoO] and are publicly available[89].

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

## Acknowledgements

We are grateful to Severio Bellusci and Harold Chapman for providing *Sftpc-CreER* mice and to Claudia Scholl and Brigit Hogan for *Scgb1a1-CreER* and *Foxj1-CreER* mice. We thank Simone Kraut, Marion Bähr, the DKFZ Core Facilities of Light Microscopy, Flow Cytometry, Small Animal Imaging Center and Genomics and Proteomics for the excellent technical assistance; and the Central Animal Laboratory for animal husbandry. We appreciate the help of Jan-Philipp Mallm and the DKFZ Single-Cell Sequencing Open Lab in designing and conducting the scRNA-seq experiments. We wish to thank Alberto Diaz, Alicia Alonso, Maria Ramos and Kalman Somogyi for their suggestions on the manuscript. Schemes have been generated with BioRender.com. This work was supported by the Deutsches Zentrum für Lungenforschung (DZL, German Center for Lung Research # 82DZL004A4) to R.S. and C.P.; Y.C. and D.W. were supported by the Helmholtz Foundation. G.T.S. was supported by the Graduate College (Graduiertenkolleg, GRK) #2338 of the German Research Society (Deutsche Forschungsgemeinschaft, DFG), the target validation project for pharmaceutical development ALTERNATIVE of the German Ministry for Education and Research (Bundesministerium für Bildung und Forschung, BMBF) and a Translational Research Grant by the German Centre for Lung Research (DZL).

## Author contributions

Y.C., G.T.S., S.C., C.P. and R.S. designed the experiments. Y.C. and S.C. performed the experiments and analysed the data. R.T. and J.H. conducted bioinformatic analyses. R.T. and P.L. developed bioinformatic methods. Y.C. and D.W. performed the TBWGBS. S.S. developed the method to detect mouse lung tumours by μCT. Y.C., S.C., R.T., C.P. and R.S. wrote the manuscript with comments from all authors.

## Funding

## Competing interests

The authors declare no competing interests.
