## [Peer Review File · Nature Communications]

Club cells employ regeneration mechanisms during lung tumorigenesisThis manuscript has been previously reviewed at another journal that is not operating a transparent peer review scheme. This document only contains reviewer comments and rebuttal letters for versions considered at *Nature Communications*.

Reviewers' comments:

Reviewer #1 (Remarks to the Author): expertise in lineage tracing studies from single cell RNA-seq data

Review comment

In this manuscript entitled 'Diverse routes of Club cell evolution in lung adenocarcinoma', Chen, Toth, and Chocarro et al. explored mouse lung cancer model induced of Eml4-Alk fusion by CRISPR/Cas9 system (previously developed by Maddalo et al.). The authors proclaimed that club cells are the origin of this type of lung cancer by lineage tracing with cell type specific GFP labeling. They also claim that club cells lose their lineage fidelity by epigenetic mechanisms during transformation and eventually take up an alveolar type 2 (AT2) like phenotype. The study is intriguing in that identifying club cells as the cell-of-origin of ALK rearranged lung cancer and suggesting an epigenetic mechanism transforming club cells into AT-2 like phenotype. In this study design, however, Scgb1a1 line (used for club cell specific GFP positivity) label both club cells and AT2 cells. Therefore, no conclusions can be drawn regarding the tumor cell of origin. Genetic engineering of cell type specific manner seems to be a better strategy to prove club cells as the origin of Eml4-Alk fusion.

(Suggestion) Tumor cell alignment, with the data in Marjanovic et al. (2020) may well demonstrate systemic differences among the different oncogenic models and clues on the origin of tumor cells.

Specific comments

1. Figure 2: As Scgb1a1 line labels both club cells and AT2 cells, GFP positivity does not prove club cell of origin in EML4-Alk tumor. The higher rate of GFP+ lesions in the Scgb1a1 than Sftpc line is only suggestive.
2. Figure 2D_Foxj1 panel: There is no background GFP signal labeling normal ciliated cells. Does it mean that the GFP labeling did not work for some mice?
3. Club cell origin of tumor is unique in Eml4-Alk? What about other lung cancer model? If Kras G12D is used instead of Eml4-Alk for the transformation, only AT2 cells emerge as the cell-of-origin? If testing Kras model is not durable, can you check public single cell RNA sequencing data whether early stage Kras tumor express club cell transcripts?
4. Line 217-221: GFP+ cells from the tumor region contain some normal cells, which contaminate the tumor cell signals in bulk methylome analysis. Were the tumor samples cleaned up of normal cell contaminants before TWGBS experiments? Otherwise, the signal may be remnants of contaminating normal cells.
5. Lin 224-225: Figure 4 demonstrates all tumors have AT2-like methylation patterns. No evidence of lineage switch during transformation is presented.
6. In the single cell RNAseq experiment: what is the proportion of AT2 cells among GFP+ cells in the tumor induction groups/time points?
7. Line299: Lcn2 is not an AT2 identity gene. Isn't the expression highest in club cell subcluster?
8. The scRNAseq populations are selected as Scgb1a1 driven GFP positive cells. Can you integrate the data to public reference to see how well your normal populations align with public reference cells?
9. Can you check tumor (cluster) identity with copy number inference?
10. Line313-314: The localization of cluster 6b in the UMAP space is not clear. Specify 6b in the cluster.
11. Figure 7A and B: Please mark positive staining. It is difficult to discern specific staining. Statistical presentation of positive signals in different time point/groups would be helpful to convince authors' claim.

Reviewer #2 (Remarks to the Author): expertise in bioinformatics and epigenetics, particularly RNA-Velocity

The current study focuses on identifying the cell of origin for ALK rearranged lung tumors. They specifically employ genetically engineered mice models (GEMMs) to infer the lineage of cells giving rise to lung cancer induced using an adenovirus delivery of a CRISPR/Cas9 construct targeting the

Eml4 and Alk genes. The authors first assess the relevance of this adenovirus-CRISPR approach to generate ALK rearranged lung tumours in mice. They next use this approach on four GEMMs, each designed to label different lung cells based on "cell-specific" guided expression of CreER. Imaging and DNA methylation of sorted labelled cells captured from these GEMMs was then used to infer the cell of origin of tumours. Next, they used scRNAseq across different time points in the scgb1a1-CreER model meant to label Club cells to further assess the likely cell of origin. Using RNA-Velocity, the authors demonstrate two potential routes of oncogenesis, including one resulting in an AT2-like state in models meant to be originating from Club cells. The authors conclude that ALK rearranged lung tumors originate from Club cells that can engage in one of two oncogenic paths.

Concerns:

As stated by the authors, the GEMM models used are leaky and do not provide convincing evidence for the cell of origin to ALK rearranged lung tumors.

The results are correlative and lack functional assessment of the role of Club cells towards ALK rearranged lung cancer

Reviewer #3 (Remarks to the Author): expertise in mouse models of lung adenocarcinoma

Chen and colleagues address the cell of origin of Eml4-Alk rearranged driven lung adenocarcinoma. By using elegant genetic cell line tracing experiments, they identify the Club cells as the most likely cell-of-origin in Eml4-Alk mutated LUAD. Additionally, and in agreement with previous reports, they identify a cell lineage switch from Club cells toward AT2-like cells during tumorigenesis, and delineate two major paths of progression from Club cells to tumor cells.

The manuscript addresses an important question on the field and it is of interest.

Major concerns

1) Although the authors demonstrate that the adenoviral system can infect most/all lung epithelial cell types, they formally do not show that the activity of the Cas9/sgRNA is equal in all the cells. Cas9 could be more efficient in inducing the Eml4-Alk rearrangement in Club cells compared to AT2 cells and therefore explain the higher tumor numbers derived from mice harboring Club cells GFP labelled. This should be addressed, ideally in vivo, but at least in vitro by isolating eGFP positive cells from the AT2 and Club cells reporter tamoxifen induced mice, transducing them with Ad-EA and checking the Eml4-Alk rearrangement efficiency.

2) A control that seems to be missing in the manuscript is the efficiency of activation of the reporter mice. If the Club cell reporter mouse hit significant more cells than the AT2 reporter, the increased number of GFP tumors in the Club cell reporter mouse can be just due to an increase of GFP cells that initially can be targeted by the Ad-EA. What are the GFP/corresponding cell marker (double and single positives) cell numbers of the four reporter mice after the tamoxifen treatment? Also, what are the percentages/numbers of labelled AT2 and Ciliated cells in the Scgb1a1 CreERT line? In Kathiriyala et al 2020, the number of SPC+ cells are considerable (33%). Depending on the absolute labelled cell numbers it could impact on the cell tracing experiments. This should be carefully addressed.

3) If Club cells are mainly the cell of origin of Eml4-Alk rearrangement LUAD, this would implicate that they are "easier to transform" compared to the AT2 cells. This could be tested by isolating GFP labelled Club and AT2 cells, transducing them with Ad-EA and analyzing if Club cells show more signs of cellular transformation, i.e. cell proliferation, perhaps even immortalization? or changes in global gene expression that reflects a more transformed phenotype in Club cells compared to AT2 cells.

4) If I am not mistaken, the analysis of human TCGA lung tumors was done without selecting Eml4-Alk rearrange tumors. If so, I am not sure how significant are the results of the authors, since this may simply show that the Club cell-derived tumors gene signatures having prognosis value are "general" tumor gene signatures (i.e. they would confer prognosis independent of the

genetic driver or maybe even of the tumor type). I agree with the authors that this validates some of the gene signatures from the murine Club cell-derived tumors in human tumors, but it does not really tell too much about the role of Club cells in human tumors. I would be more interested in seeing if Club cells-derived tumors gene expression signatures match better human LUAD (ideally mutated Eml4-Alk) than AT2 cells-derived tumors expression signatures. This may support more the idea that Club cells are also the cell of origin in human Eml4-Alk (or other genetic drivers) mutated LUAD.

5) Figure 2D, how many mice were used in these experiments? In general, the figure legends lack of detailed information. This should be improved across the manuscript. Also a figure showing the GFP/tomato percentage of tumors of individual animals should be included. The SE, SD or SEM (it seems not defined) in the Sftpc and in the Hoxp groups seems to be quite large ($12\% \pm 9$ and $19\% \pm 14$, respectively). Are the differences between the four groups statistically significant?

6) Rather than probe for "total" nuclear STAT1, it would be more correct to perform the stainings with activated pSTAT1. Please, include pictures with higher magnifications and indicate with arrows the nuclear pSTAT1/STAT1. From the pictures, it is difficult to see if it is really nuclear or in the whole cell.

Minor point

In general, it is more accepted that the cell of origin in mutated KRAS and EGFR mouse models of LUAD are the AT2 cells. I wonder if this is due to the "forced" cell specific activation of the oncogenes in a particular cell type used in previous models, in contrast the more "unbiased" approach used by the authors, or if it is due to the use of the Eml4-Alk diver. This could be included in the discussion.

Point-by-point rebuttal

Reviewers' comments:

Reviewer #1 (Remarks to the Author): expertise in lineage tracing studies from single cell RNA-seq data

Review comment

In this manuscript entitled 'Diverse routes of Club cell evolution in lung adenocarcinoma', Chen, Toth, and Chocarro et al. explored mouse lung cancer model induced of Eml4-Alk fusion by CRISPR/Cas9 system (previously developed by Maddalo et al.). The authors proclaimed that club cells are the origin of this type of lung cancer by lineage tracing with cell type specific GFP labeling. They also claim that club cells lose their lineage fidelity by epigenetic mechanisms during transformation and eventually take up an alveolar type 2 (AT2) like phenotype. The study is intriguing in that identifying club cells as the cell-of-origin of ALK rearranged lung cancer and suggesting an epigenetic mechanism transforming club cells into AT-2 like phenotype. In this study design, however, Scgb1a1 line (used for club cell specific GFP positivity) label both club cells and AT2 cells. Therefore, no conclusions can be drawn regarding the tumor cell of origin. Genetic engineering of cell type specific manner seems to be a better strategy to prove club cells as the origin of Eml4-Alk fusion.

(Suggestion) Tumor cell alignment, with the data in Marjanovic et al. (2020) may well demonstrate systemic differences among the different oncogenic models and clues on the origin of tumor cells.

This is a very good suggestion from the reviewer. We considered this upon the preparation of our manuscript, however, we decided against it. The reason being that the fitting data from Marjanovic et al. *Cancer Cell* 2020, is different from the one presented here; not only it captures a different cell type composition, but also there are substantial technical differences (different protocol, cell number and sequencing depth) which makes the integration challenging. Therefore, the results might not be conclusive, as it is difficult to judge how successful the integration was.

However, the bioinformatic methods for scRNA sequencing develop fast, and we will try the integration with recently developed tools (e.g. Hao Y et.al., *Cell* 2021).

Specific comments

1. Figure 2: As Scgb1a1 line labels both club cells and AT2 cells, GFP positivity does not prove club cell of origin in EML4-Alk tumor. The higher rate of GFP+ lesions in the Scgb1a1 than Sftpc line is only suggestive.

We agree with the reviewer that *Scgb1a1-Cre-ER* mice, although have been extensively used in the literature to mark Club cells, they cannot prove a definitive Club cell origin of the tumors. This is the reason why we did methylome analysis in addition of the lineage tracing.

In addition, we observed 19% of GFP positive tumors in the *Hopx* line and since this line labels 35% of club cells, we can confirm the club origin of the tumors.

2. Figure 2D_Foxj1 panel: There is no background GFP signal labeling normal Ciliated cells. Does it mean that the GFP labeling did not work for some mice?

Foundation under Public Law

Management Board
Prof. Dr. med. Michael Baumann
Ursula Weyrich

Deutsche Bank Heidelberg
IBAN: DE09 6727 0003 0015 7008 00
BIC (SWIFT): DEUT DES M672

Deutsche Bundesbank Karlsruhe
IBAN: DE39 6600 0000 0067 0019 02
BIC (SWIFT): MARK DEF 1660

The labeling did work on this mouse, but it is true that with the scale used it is difficult to see. We will include a zoom image showing the Ciliated cells labeled in this image. In any case, the specific labeling of *Foxj1* can be seen in figure 2A.

3. Club cell origin of tumor is unique in Eml4-Alk? What about other lung cancer model? If *Kras* G12D is used instead of Eml4-Alk for the transformation, only AT2 cells emerge as the cell-of-origin?

If testing *Kras* model is not durable, can you check public single cell RNA sequencing data whether early stage *Kras* tumor express club cell transcripts?

This is a very interesting and important question from the reviewer, whether Club cells are the cells of origin of other mutant lung adenocarcinomas. In fact, for a long time it was believed that AT2 cells were the origin of mutant *Kras* lung adenocarcinomas (Kim et al., *Cell* 2005; Sutherland et al., *PNAS* 2014; Xu et al., *PNAS* 2012), however, two recent papers (Concepcion C et al., *Cancer Discovery* 2021, and Rosigkeit et al., *IJC* 2021) show that Club cells also originate *Kras* mutant tumors. Unfortunately, these two papers were published after our submission, but we will include them both in the introduction as well as in the discussion.

We appreciate the reviewer's suggestion of testing our hypothesis in a different lung cancer model, such as *Kras* G12D, however in light of the recent publications we believe that the results showing that Club cells can originate other oncogene induced lung cancer have already been shown, although the mechanism by which this happens has never been determined.

4. Line 217-221: GFP+ cells from the tumor region contain some normal cells, which contaminate the tumor cell signals in bulk methylome analysis. Were the tumor samples cleaned up of normal cell contaminants before TWGBS experiments? Otherwise, the signal may be remnants of contaminating normal cells.

We believe this possibility is very unlikely, since tumor nodules were macroscopically removed and GFP+ cells were sorted. Based on the scRNASeq analysis, only a very small number of cells are showing expression of *Scgb1a1* (Appeal Figure 1 is showing only the cells from the Tumor sample in the scRNAseq experiment and the colors reflect the *Scgb1a1* expression). This number of cells would not show a strong signal in the bulk methylome analysis.

Appeal Figure 1 Expression of Scgb1a1 in the cells of the scRNAseq Tumor sample

Also, we see the Club signature from the tumors arising from the *Hopx* model. If the signal was coming from normal cells, here we would see a lower level of LMC1 (capturing the Club signature) and higher level from LMC3 (capturing AT2 and AT1 cells, see Figure 4C). In Figure 4E, the region highlighted in blue gives further evidence against normal contamination. This region shows normal Club cell pattern in the tumors. If there was a significant contamination of normal cells, we should see an intermediate level of methylation here, which is closer to those in AT2 cells.

This discussion can be included in the paper.

5. Lin 224-225: Figure 4 demonstrates all tumors have AT2-like methylation patterns. No evidence of lineage switch during transformation is presented.

We disagree with this statement. We do agree that Figure 4 shows that all tumors have an AT2 methylation pattern. In figure 4D, we clearly show that tumors arising from the *Hopx* line (that can only originate from Club, Ciliated or AT1 cells) have a similar methylation pattern as those tumors arising from *Scgb1a1*. And all together, these “Club originated tumors” show the *Sftpc* pattern arising after the lineage switch, although they retain some epigenetic marks similar to normal Club cells.

In addition, Supplementary Figure 1A shows that all tumors are SPC positive. However, supplementary Figure 3C, clearly shows a tumor growing inside a bronquios, which is labeled in *Scgb1a1* and is SPC positive, suggesting there is a lineage switch. With Figure 4, we aimed to show the hypermethylation of the *Scgb1a1* gene in Club originated tumors, impairing the expression of this marker, and the hypomethylation of the *Sftpc* gene in these tumors as a consequence of the lineage switch.

In Figure 4F, the methylation pattern of transcription factor binding motifs gives us further evidence for the lineage switch. We see that the hypermethylation pattern of Club originated tumorigenesis (*Scgb1a1* N vs. T) affects those TFs that are also differentially methylated between the cell types (*Scgb1a1* T vs. *Sftpc* T), while the hypomethylation patterns are showing only tumorigenesis related changes.

6. In the single cell RNAseq experiment: what is the proportion of AT2 cells among GFP+ cells in the tumor induction groups/time points?

In Figure 5D we show the ratio of cells found in Cluster 7 (identified as AT2 cells) from each sample. In addition, in Figure 5H we show the cell type composition of each sample, expressed as the ratio of each cell type in each sample. Briefly, 61.02% of the cells in Cluster 7 come from the 4wk-GFP sample, in which AT2 cells were not sorted out. As for the other samples, the % of AT2 cells are:

TAM: 1.30%

Cas9: 2.79%

2 wk-1 : 0.89%

2 wk-2 3.37%

4 wk-1: 6.75%

Tumour: 5.96%

We have included this data in Supplementary table 6.

7. Line299: *Lcn2* is not an AT2 identity gene. Isn't the expression highest in club cell subcluster?

Lcn2 is not an AT2 identity gene, but several publications define this gene as a marker of activated or primed AT2s (Strunz et al., *Nat Common* 2020, Kobayash et al., *Nat Cell Bio* 2020). This gene appears to be a marker of our Stem-like tumor module, which claim that it is not only characterized by stem cell markers, but also AT2 markers, most probably primed AT2s. In Figure 6B we show the levels of this gene in every activity program, and indeed the Club programs is one of the modules with the lowest expression of this gene.

8. The scRNAseq populations are selected as Scgb1a1 driven GFP positive cells. Can you integrate the data to public reference to see how well your normal populations align with public reference cells?

We will integrate our data with the Adult Lung cells from the Mouse Cell Atlas (Han X et al., *Cell* 2018) and with the Aging Lung Atlas (Angelidis I et al., *Nature Communications* 2019) and if the results will allow, include them in the paper.

9. Can you check tumor (cluster) identity with copy number inference?

To assess the tumor identity, we counted the reads coming from the translocated part of the Alk gene, that, as an oncogene is not expressed in normal cells. We agree with the Reviewer that adding the copy number inference can support this finding, therefore we will include this in the paper.

10. Line313-314: The localization of cluster 6b in the UMAP space is not clear. Specify 6b in the cluster.

We apologize for not showing this clearly. This has been changed.

11. Figure 7A and B: Please mark positive staining. It is difficult to discern specific staining. Statistical presentation of positive signals in different time point/groups would be helpful to convince authors' claim.

We can do a quantification of the positive staining in each time point. In addition, as suggested by another reviewer we will perform pStat1 staining.

Reviewer #2 (Remarks to the Author): expertise in bioinformatics and epigenetics, particularly RNA-Velocity

The current study focuses on identifying the cell of origin for ALK rearranged lung tumors. They specifically employ genetically engineered mice models (GEMMs) to infer the lineage of cells giving rise to lung cancer induced using an adenovirus delivery of a CRISPR/Cas9 construct targeting the Eml4 and Alk genes. The authors first assess the relevance of this adenovirus-CRISPR approach to generate ALK rearranged lung tumours in mice. They next use this approach on four GEMMs, each designed to label different lung cells based on "cell-specific" guided expression of CreER. Imaging and DNA methylation of sorted labelled cells captured from these GEMMs was then used to infer the cell of origin of tumours. Next, they used scRNAseq across different time points in the scgb1a1-CreER model meant to labelled Club cells to further assess the likely cell of origin. Using RNA-Velocity, the authors demonstrate two potential routes of oncogenesis, including one resulting in an AT2-like states in models meant to be originating from Club cells. The authors conclude that ALK rearranged lung tumors originate from Club cells that can engage in one of two oncogenic paths. Concerns:

As stated by the authors, the GEMM models used are leaky and do not provide convincing evidence for the cell of origin to ALK rearranged lung tumors.

We thank this reviewer for taking the time in reading our manuscript.

We disagree that all GEMMs used in the study are leaky. The *Scgb1a1* line marks Club and also AT2 cells, however the *Sftpc* line is not leaky and marks exclusively AT2 cells.

Although the *Scgb1a1* line can label AT2 cells, it has been widely used by several researchers in many publications, Kathiriya et al., *Cell Stem Cell* 2020; Rosigkeit et al., *IJC* 2021; Liu et al., *Nat Genetics* 2019; Choi et al., *Nat Cell Bio* 2021; Pardo-Saganta et al., *Nature* 2015; McConnell et al., *Cell Reports* 2016; Rawlins et al., *Cell Stem Cell* 2009; etc to label Club cells.

To our knowledge and also to reviewers #1 and #3 these are widely used and elegant GEMMs.

The results are correlative and lack functional assessment of the role of Club cells towards ALK rearranged lung cancer

As suggested by another reviewer we will now perform additional experiments to assess the role of Club and AT2 cells in Alk driven lung cancer. We will sort CD45⁻/ CD31⁻/EpCAM⁺/ tdTomato⁻/ CD24⁻/ β 4⁺/ CD200⁺ Club cells and CD45⁻/ CD31⁻/EpCAM⁺/ tdTomato⁻/ CD24⁻/ β 4⁻/ CD200⁺ AT2 cells and transduce them with Ad-AE in vitro. We will analyze signs of cellular transformation, such as proliferation potential, immortalization and changes in gene expression.

Reviewer #3 (Remarks to the Author): expertise in mouse models of lung adenocarcinoma

Chen and colleagues address the cell of origin of Eml4-Alk rearranged driven lung adenocarcinoma. By using elegant genetic cell line tracing experiments, they identify the Club cells as the most likely cell-of-origin in Eml4-Alk mutated LUAD. Additionally, and in agreement with previous reports, they identify a cell lineage switch from Club cells toward AT2-like cells during tumorigenesis, and delineate two major paths of progression from Club cells to tumor cells.

The manuscript addresses an important question on the field and it is of interest.

Major concerns

1) Although the authors demonstrate that the adenoviral system can infect most/all lung epithelial cell types, they formally do not show that the activity of the Cas9/sgRNA is equal in all the cells. Cas9 could be more efficient in inducing the Eml4-Alk rearrangement in Club cells compared to AT2 cells and therefore explain the higher tumor numbers derived from mice harboring Club cells GFP labelled. This should be addressed, ideally in vivo, but at least in vitro by isolating eGFP positive cells from the AT2 and Club cells reporter tamoxifen induced mice, transducing them with Ad-EA and checking the Eml4-Alk rearrangement efficiency.

This is a very interesting point. We have performed FISH analysis to detect the rearrangement in lung sections from infected mice. However, these data were not conclusive as the number of cells containing the rearrangement at early time points is very low. Later time points would not be conclusive as we show that Club transformed cells quickly become AT2 like.

For all these reasons, we will follow the reviewer's suggestion and will transduce Club and AT2 cells grown in 3D organoids and check the rearrangement efficiency.

2) A control that seems to be missing in the manuscript is the efficiency of activation of the reporter mice. If the Club cell reporter mouse hit significant more cells than the AT2 reporter, the increased number of GFP tumors in the Club cell reporter mouse can be just due to an increase of GFP cells that initially can be targeted by the Ad-EA.

What are the GFP/corresponding cell marker (double and single positives) cell numbers of the four reporter mice after the tamoxifen treatment? Also, what are the percentages/numbers of labelled AT2

and Ciliated cells in the Scgb1a1 CreErt line? In Kathiriya et al 2020, the number of SPC+ cells are considerable (33%). Depending on the absolute labelled cell numbers it could impact on the cell tracing experiments. This should be carefully addressed.

As stated by the reviewer, these are very important controls. We did not include them in the manuscript since all these mouse models have been extensively used in past publications. However, we had performed all this analysis in our initial experiments to characterized the mouse models ourselves. We have included all this data in a new table, revised Supplementary Table 1 on the revised manuscript.

In our analysis, the **Scgb1a1-CreER** mice after tamoxifen labels 61% of Club cells, 11% of the AT2 population and 23% of Ciliated cells.

The **Sftpc-CreER** line labels 45% of the AT2 population after 4 weeks of tamoxifen induction. We found no other cell type labeled in these animals.

The other mouse line used in this study is **Hopx-CreER**. In this case, we found 35% of Club cells labeled, 13% of AT1 cells and 19% of Ciliated cells.

Foxj1 animals labeled only Ciliated cells (28%) after 4 weeks of tamoxifen and 1% of Club cells.

We believe that due to the much lower amount of AT2 cells labeled in the Scgb1a1 line compared to the Sftpc, the higher percentage of GFP+ tumors in Scgb1a1 mice could not be due to AT2 origin of the tumors. In addition, 61% of the Club cells are labeled in the Scgb1a1 model, while only 19% are labeled in the Hopx line. This data correlate with the lower percentage of GFP+ tumors in the Hopx mice (19%) compared to the Scgb1a1 (44% of green tumors). We have included these tables in a new Supplementary Table 1.

		AT2 cells			Club cells			Ciliated cells		
Sample		SPC	GFP-SPC	labelling	CCSP	GFP-CCSP	labelling	acTub	GFP-acTub	labelling
Scgb1a1	GCRC504	31861	3520	11,05	123012	95783	77,86	29646	4241	14,31
	GCRC507	23334	1497	6,42	8018	3947	49,22	816	218	26,73
	GCRC514	26120	4086	15,64	40990	17430	42,52	5716	968	16,93
	GCRC517	16450	2051	12,47	166537	126740	76,10	6320	2247	35,56
	Average			11,39			61,43			23,38
SD			3,32			15,75			8,42	

		AT2 cells		
Sample		SPC	GFP-SPC	labelling
Sftc	DTS118	72521	42914	59,17
	DTS120	35361	17883	50,57
	GCRS506	53655	18815	35,07
	GCRS508	37494	13865	36,98
Average			45,45	
SD			9,93	

		AT1 cells			Club cells			Ciliated cells		
Sample		PDPN	GFP-PDPN	labelling	CCSP	GFP-CCSP	labelling	acTub	GFP-acTub	labelling
Hopx	GCRH425	14576179	2186621	15,00	67417	21807	32,35	17109	4201	24,56

	GCRH426	12481118	1955346	15,67	22142	8639	39,02	5015	848	16,90
	GCRH433	13583189	1397243	10,29	57924	20703	35,74	34860	5611	16,10
	Average			13,65			35,70			19,18
	SD			2,39			2,72			3,81

		Ciliated cells			Club cells		
Sample		acTub	GFP+acTub	labelling	CCSP	GFP+CCSP	labelling
Foxj1	GCRF9	7368	2956	40,12	7272	96	1,32
	GCRF10	11729,09	1919	16,36	7758	88	1,14
	Average			28,24			1,23
	SD			11,88			0,09

3) If Club cells are mainly the cell of origin of Eml4-Alk rearrangement LUAD, this would implicate that they are “easier to transform” compared to the AT2 cells. This could be tested by isolating GFP labelled Club and AT2 cells, transducing them with Ad-EA and analyzing if Club cells show more signs of cellular transformation, i.e. cell proliferation, perhaps even immortalization? or changes in global gene expression that reflects a more transformed phenotype in Club cells compared to AT2 cells.

We agree with the reviewer and this is a very important point. We will sort CD45⁻/ CD31⁻/EpCAM⁺/ tdTomato⁻/ CD24⁻/ β4⁺/ CD200⁺ Club cells and CD45⁻/ CD31⁻/EpCAM⁺/ tdTomato⁻/ CD24⁻/ β4⁻/ CD200⁺ AT2 cells and transduce them with Ad-AE. We will characterize those cells according to the reviewer’s suggestion.

4) If I am not mistaken, the analysis of human TCGA lung tumors was done without selecting Eml4-Alk rearranged tumors.

If so, I am not sure how significant are the results of the authors, since this may simply show that the Club cell-derived tumors gene signatures having prognosis value are “general” tumor gene signatures (i.e. they would confer prognosis independent of the genetic driver or maybe even of the tumor type). I agree with the authors that this validates some of the gene signatures from the murine Club cell-derived tumors in human tumors, but it does not really tell too much about the role of Club cells in human tumors. I would be more interested in seeing if Club cells-derived tumors gene expression signatures match better human LUAD (ideally mutated Eml4-Alk) than AT2 cells-derived tumors expression signatures. This may support more the idea that Club cells are also the cell of origin in human Eml4-Alk (or other genetic drivers) mutated LUAD.

Yes, this analysis was done using all the samples from TCGA, regardless of the driver alteration, as the number of EML4-ALK rearranged LUAD samples was very low.

We apologize for the misunderstanding, as this analysis was not intended to investigate the role of Club cells in human tumors. Our goal was to show that the signatures identified in our different tumor clusters are present and might have different effects in human LUAD.

We can do additional investigations comparing Club and AT2 tumor expression signatures, depending on the result of the integration of our data with the data from Marjanovic et al. *Cancer Cell* 2020 (AT2 gene expression signatures). The integration of these data has also been suggested by Reviewer 1.

5) Figure 2D, how many mice were used in these experiments? In general, the figure legends lack of detailed information. This should be improved across the manuscript. Also a figure showing the GFP/tomato percentage of tumors of individual animals should be included. The SE, SD or SEM (it seems not defined) in the *Sftpc* and in the *Hoxp* groups seems to be quite large (12%±9 and 19%±14, respectively). Are the differences between the four groups statistically significant?

We apologize for the lack of details in our figure legends. We have improved them as the reviewer suggested. In Figure 2D and new revised Figure 2D and E; the numbers are as follow:

	mouse id	GFP+ tumors	total tumors	% GFP
Scgb1a1	GCRC21	29	63	46,03
	GCRC79	10	27	37,04
	GCRC31	20	41	48,78
	mean			43,95
			sd	5,02

	mouse id	GFP+ tumors	total tumors	% GFP
Sftpc	GCRS39	2	32	6,25
	GCRS145	3	22	13,64
	GCRS125	18	93	19,35
	GCRS101	11	40	27,50
	GCRS12	0	7	0,00
	GCRS40	1	20	5,00
	mean			11,96
			sd	9,34

	mouse id	GFP+ tumors	total tumors	% GFP
Hoxp	CRH83	8	44	18,18
	GCRH417	14	38	36,84
	GCRH419	1	35	2,86
	mean			19,29
			sd	13,90

	mouse id	GFP+ tumors	total tumors	% GFP
Foxj1	GCRF17	0	56	0,00
	GCRF102	0	27	0,00
	GCRF111	0	14	0,00
	GCRF119	0	16	0,00

We have also included this data in a revised Supplementary Table 2. We have performed statistical analysis, and the *Scgb1a1* line is statistically higher compared to the other 3 groups. There are no significant differences among the other groups (Appeal Figure 2 and revised Figure 2D).

Appeal Figure 2. Percentage of GFP positive tumors in each of the lineage tracing mouse lines.

6) Rather than probe for “total” nuclear STAT1, it would be more correct to perform the stainings with activated pSTAT1. Please, include pictures with higher magnifications and indicate with arrows the nuclear pSTAT1/STAT1. From the pictures, it is difficult to see if it is really nuclear or in the whole cell.

This is a very nice recommendation. We will perform this stainings as suggested.

Minor point

In general, it is more accepted that the cell of origin in mutated KRAS and EGFR mouse models of LUAD are the AT2 cells. I wonder if this is due to the “forced” cell specific activation of the oncogenes in a particular cell type used in previous models, in contrast the more “unbiased “ approach used by the authors, or if it is due to the use of the Eml4-Alk diver. This could be included in the discussion.

We fully agree with this reviewer and we do mention this point in the introduction. As suggested above we have included this point together with the new published papers (Concepcion C et al., *Cancer Discovery* 2021, and Rosigkeit et al., *IJC* 2021) in which they show that Club cells also originate *Kras* mutant tumors.

Reviewers' comments:

Reviewer #1 (Remarks to the Author): expertise in lineage tracing studies from single cell RNA-seq data

Reviewer #1

[In this manuscript entitled „Diverse routes of Club cell evolution in lung adenocarcinoma“, Chen, Toth, and Chocarro et al. explored mouse lung cancer model induced of Eml4-Alk fusion by CRISPR/Cas9 system (previously developed by Maddalo et al.). The authors proclaimed that club cells are the origin of this type of lung cancer by lineage tracing with cell type specific GFP labeling. They also claim that club cells lose their lineage fidelity by epigenetic mechanisms during transformation and eventually take up an alveolar type 2 (AT2) like phenotype. The study is intriguing in that identifying club cells as the cell-of-origin of ALK rearranged lung cancer and suggesting an epigenetic mechanism transforming club cells into AT-2 like phenotype. In this study design, however, Scgb1a1 line (used for club cell specific GFP positivity) label both club cells and AT2 cells. Therefore, no conclusions can be drawn regarding the tumor cell of origin. Genetic engineering of cell type specific manner seems to be a better strategy to prove club cells as the origin of Eml4-Alk fusion.

Tumor cell alignment, with the data in Marjanovic et al. (2020) may well demonstrate systemic differences among the different oncogenic models and clues on the origin of tumor cells.]

[author rebuttal.....]

(To the rebuttal) It is difficult to follow the “Appeal Figure 1”, and characteristics of the integrated cluster are not presented. One clear observation is that integration of the two batches (current study and Marjanovic’s) is poor. I agree with the authors that integration, in its current form, is confusing and difficult to follow.

[Specific comments

1. Figure 2: As Scgb1a1 line labels both club cells and AT2 cells, GFP positivity does not prove club cell of origin in EML4-Alk tumor. The higher rate of GFP+ lesions in the Scgb1a1 than Sftpc line is only suggestive.

Author rebuttal: We agree with the reviewer that Scgb1a1-Cre-ER mice, although it has been extensively used in the literature to mark Club cells, cannot prove a definitive Club cell origin of the tumours. This is the reason why we did methylome analysis in addition of the lineage tracing. Moreover, we observed 19% of GFP positive tumours in the Hopx line and since this line labels 35% of club cells, it confirms the Club origin of these tumours.

We would like to clarify to this reviewer that our intention was not to claim that Club cells are the only or main origin of Eml4-Alk tumours. As described in the manuscript both Club- as well as AT2-cells can give rise to tumours. In an effort to further prove that Club cells can be the cell of origin, during the course of the revision, we performed an experiment in which we significantly reduced the amount of Tamoxifen (TAM) given to the Scgb1a1 mice (1mg) as recently published by Rosigkeit et al., 2021. This dose of TAM only labels Club cells, and here we show that we continue to observe green tumours (Appeal figure 2).]

(To the rebuttal) The specific comment 1 is the major criticism in this study. Based on the publications by Concepcion and by Rosigkeit, lung adenocarcinoma may originate from club cells (oncogenic KRAS driven tumor compared to Eml4-Alk fusion in this study). Although I understand the authors’ points on numbers, transformation in 11% AT2 cells can over-ride 61% Club cells in Scgb1a1 mice. Although authors showed 35% unspecifically labeled club cells in Hopx line, the model was generated to label AT1 cells. These two facts put weakness in the current study, and make study flow complicated.

As the authors commented, data from the low dose TAM which only labels club cells would have been more convincing.

(To the rebuttal to Specific comment #9) Without positive samples, negative data (lack of CNV) is not convincing. This addition may be left out or presented with positives.

Other comments have been addressed in satisfaction.

Reviewer #2 (Remarks to the Author): expertise in bioinformatics and epigenetics, particularly RNA-Velocity

My previous comments were not convincingly addressed. I still remain concerned about the "cell of origin" claimed in this manuscript based on the GEMM models employed and the lack of direct assessment of the functional role of Club cells towards ALK-rearranged lung cancer.

Reviewer #3 (Remarks to the Author): expertise in mouse models of lung adenocarcinoma

Dear Authors, yes indeed it has been a misunderstanding. Now, it is clear that the Authors propose Club and AT2 cells as cells-of-origin of EML Eml4-Alk induced LUAD.

I appreciate that the authors tried to address all my concerns, although most of my points were trying to clarify if Club cells were the main cell-of-origin of LUAD. I agree with the authors that the careful characterization of the reporter mouse models, and the new experiment using a low dose tamoxifen, strongly suggest that Club cells contribute to Eml4-Alk mutated LUAD. I would suggest to include the results of the Scgb1a1 mice treated with one single dose of tamoxifen in the manuscript, ideally with a quantification of GFP/cells markers as done in the new supplementary table 1.

Overall, the manuscript has improved and I am happy to recommend its acceptance.

Reviewers' comments:

Reviewer #1

Tumor cell alignment, with the data in Marjanovic et al. (2020) may well demonstrate systemic differences among the different oncogenic models and clues on the origin of tumor cells.

(To the rebuttal) It is difficult to follow the "Appeal Figure 1", and characteristics of the integrated cluster are not presented. One clear observation is that integration of the two batches (current study and Marjanovic's) is poor. I agree with the authors that integration, in its current form, is confusing and difficult to follow.

We agree with the reviewer that the integration is poor due to the differences in sequencing methods used by Marjanovic and our study. Last month, a new paper was published in which AT2 derived *Kras* mutant; *p53* null tumours were sequenced using 10X scRNA-seq (Yang et al., Cell 2022).

Therefore, we decided to integrate our data with Yang's et al. data and exchanged the results in the manuscript. This new integration clearly shows that our Club-derived tumour cell clusters separate from their AT2 derived tumour cells (Revised Figures 7D, S7G and S7H). We also investigated how the gene expression module scores of our tumour clusters are distributed among our clusters and the ones from Yang et al. (Revised Figure 7E). We found that both the AT1-like tumour and the Regeneration-like tumour modules were significantly higher in our clusters and were most similar to the AT1 like and Late Gastric clusters from Yang et al.

We hope this reviewer agrees now that this integration looks much better and important conclusions can be taken, pointing that our Club tumour clusters do not overlap with the AT2 ones.

Specific comments

1. Figure 2: As *Scgb1a1* line labels both club cells and AT2 cells, GFP positivity does not prove club cell of origin in EML4-*Alk* tumor. The higher rate of GFP+ lesions in the *Scgb1a1* than *Sftpc* line is only suggestive.

(To the rebuttal) The specific comment 1 is the major criticism in this study. Based on the publications by Concepcion and by Rosigkeit, lung adenocarcinoma may originate from club cells (oncogenic *KRAS* driven tumor compared to EML4-*Alk* fusion in this study). Although I understand the authors' points on numbers, transformation in 11% AT2 cells can over-ride 61% Club cells in *Scgb1a1* mice. Although authors showed 35% unspecifically labeled club cells in *Hopx* line, the model was generated to label AT1 cells. These two facts put weakness in the current study, and make study flow complicated.

As the authors commented, data from the low dose TAM which only labels club cells would have been more convincing.

We thank the reviewer for this comment; however, we disagree with the concern that because *Hopx* mice were generated to label AT1 cells, this should be used as an argument against our data. By using the *Hopx* mice we aimed to address whether AT1 cells gave rise to *Eml4-Alk* tumours, but after doing a deep characterization of these mice, we saw this strain also labels 35% of Club cells. To unveil whether the observed GFP+ tumours in the *Hopx* model were originated from AT1 or Club cells, we also used the *Krt5* mouse model. Here again, this strain was originally designed to label basal cells, but in fact, after its characterization we found that it also labelled AT1 cells. In the *Krt5* mouse model we did not observe any GFP+ tumour, suggesting that neither basal cells nor AT1 cells can give rise to *Eml4-Alk* tumours. Therefore, the GFP+ tumours in the *Hopx* mice, must be originated from Club cells and not AT1 cells.

Although transformation of 11% of AT2 cells could over-ride the 61% of Club cells, we would like to point out here that if that was the case, the percentage of GFP+ tumours in the *Sftpc* lineage tracing mice (which labels 45% of AT2 cells), should be higher than 12%. In our manuscript we use a combination of 5 different lineage tracing mouse models, *Scgb1a1*, *Sftpc*, *Hopx*, *Krt5* and *Foxj1* that all together clearly show that *Eml4-Alk* tumours can originate both from Club and AT2 cells. We also would like to clarify to this reviewer, that the percentages of GFP+ tumours are given in the manuscript as an indication and not to claim whether Club cells contribute more or less than AT2 cells to lung cancer.

We agree with the reviewer that using the *Hopx* mouse model is an indirect way of identifying the cell of origin, but the Club origin of these tumours was further confirmed by whole genome bisulfite sequencing.

In figure 4B and 4D we show that *Hopx* GFP+ tumours, similar to *Scgb1a1* GFP+ **Foundation under Public Law**

tumours, retain a Club epigenetic pattern which is different to the one from *Sftpc* GFP+ tumours.

We understand and agree with this reviewer that all these genetic lineage tracing mouse models are unfortunately promiscuous and difficult to follow, but we hope that with our deep analysis of these mouse strains, the scientific community will benefit and will be cautious in using the models.

In addition, we agree with the reviewer's comment that data from the low dose TAM would have been better, but as mentioned in our previous rebuttal, the low dose of TAM for labelling only Club cells was published shortly before our initial submission to *Nature Communications*. Nevertheless, we believe that the data provided in our rebuttal letter, showing GFP positive tumours after low dose of TAM proves that Club cells give rise to tumours.

We would like to point out and make clear to this reviewer that the main focus of our manuscript is to understand the development of Club originated tumours and not to address which cell type, if Club or AT2 gives rise to more tumours. We hope that with this explanation, reviewer #1 will now acknowledge the importance of our manuscript and will be happy to recommend its publication.

(To the rebuttal to Specific comment #9) Without positive samples, negative data (lack of CNV) is not convincing. This addition may be left out or presented with positives.

Unfortunately, we do not have positive samples that show an increase in CNVs in *Eml4-Alk* tumours. In fact, WGS of 5 selected tumour nodules from this model have no SCNAs compared to control lung (data not shown). This result is not surprising as many mouse lung tumours do not show increase copy number alterations (McFadden et. al., PNAS 2016). Moreover, in our hands we have looked at aneuploidy in a mouse model of *Kras* induced lung cancer and found very little changes (Sotillo et. al., Nature 2010). We agree with the reviewer and have removed this data from the manuscript.

Other comments have been addressed in satisfaction.

We wish to thank this reviewer and are pleased that he/she was satisfied with the rest of the comments.

Reviewer #2

My previous comments were not convincingly addressed. I still remain concerned about the "cell of origin" claimed in this manuscript based on the GEMM models employed and the lack of direct assessment of the functional role of Club cells towards ALK-rearranged lung cancer.

We agree with this reviewer that some of the GEMM models are promiscuous. This is the reason why we included in this manuscript a thorough characterization which included different controls. We even mention in the text that the solely use of the GEMMs is not sufficient to address the cell of origin. However, we would like to point out that all these models are widely used in the lung field and with this study we would also like to raise awareness of the leakiness of these models.

We hope that the argument given to reviewer #1 about the fact that we see Club-originated tumours also in the *Hopx* model will make our statement a bit clearer for reviewer #2. By using the *Hopx* mice we aimed to address whether AT1 cells gave rise to *Eml4-Alk* tumours, but after the characterization of these mice, we saw this strain also labels 35% of Club cells. To unveil whether the observed GFP+ tumours in the *Hopx* model were originated from AT1 or Club cells, we also used the *Krt5* mouse model, which labels both basal cells and AT1 cells. In the *Krt5* mouse model we did not observe any GFP+ tumour, suggesting that neither basal cells or AT1 cells can give rise to *Eml4-Alk* tumours. Therefore, the GFP+ tumours in the *Hopx* mice, must be originated from Club cells and not AT1 cells. We are aware that using the *Hopx* mouse model is an indirect way of identifying the cell of origin, but the Club origin of these tumours was further confirmed by whole genome bisulfite sequencing.

Additionally, if the 11% of labelled AT2 cells in the *Scgb1a1* mice were the ones giving rise to the GFP+ tumours, then the percentage of green nodules in the *Sftpc* mice should be higher than the 12% that we observed (where 45% of AT2 cells are labelled). In our manuscript we use a combination of 5 different lineage tracing mouse models, *Scgb1a1*, *Sftpc*, *Hopx*, *Krt5* and *Foxj1*, and although we understand these models might be confusing, we believe that all together they clearly show that *Eml4-Alk* tumours can originate both from Club and AT2 cells. We would like to clarify that the percentages of GFP+ tumours are given as an indication and not to claim if Club cells contribute more or less than AT2 cells to lung cancer.

In an effort to further prove that Club cells can be the cell of origin, during the course of the revision and as a reply to one of the questions from reviewer#1, we performed an experiment in which we significantly reduced the amount of Tamoxifen (TAM) given to the *Scgb1a1* mice (1mg) as recently published by Rosigkeit et al., 2021. This dose of TAM only labels Club cells in this model, and here we show that we continue to observe green tumours (Appeal Figure 1). We are aware that data from the low dose TAM would have been better, but the low dose of TAM for labelling only Club cells was published shortly before our initial submission to *Nature Communications*.

Appeal Figure 1. Immunofluorescent staining with GFP and RFP antibodies on lungs of *mT/mG, Scgb1a1* mice. These mice were injected with a low dose of tamoxifen (1mg), and 4 weeks later Ad-EA was administered. Only Club cells are labeled under this condition. Scale bars: 1mm.

Furthermore, we unfortunately cannot design additional experimental work with the available tools to address the functional role of Club cells towards ALK rearranged lung cancer. However, in an effort to address this reviewer's concern we have performed a few experiments which we hope will help clarify this question.

First of all, we sorted GFP+ tumour cells from *Scgb1a1* mice at humane endpoint and subcutaneously injected 1×10^5 cells into the flank of nude mice. As observed in Appeal Figure 2A, these tumour cells were able to grow and form tumours in nude mice. Additionally, to measure the proliferation rate of Club and AT2 tumours, we injected *Scgb1a1* and *Sftpc* tumour-bearing mice with BrdU, a nucleoside analogue which is incorporated during S-phase. We then measured by histology the percentage of BrdU positive cells (proliferating) in each tumour, and observed that those GFP+ tumours in *Scgb1a1* mice proliferated more than the GFP+ in *Sftpc* animals (Appeal Figure 2B), suggesting that these tumours might be more aggressive. Although this result is interesting and statistically significant, we decided not to include this data in the manuscript since we do not aim to compare tumours from the two different origins and believe this data will deviate the reader from the main message.

Appeal Figure 2

a) Volume of tumours originated from GFP+ tumor cells from *Scgb1a1* mice and subcutaneously injected into the flank of nude mice. Each curve represents a nude mouse injected with cells from an independent *Scgb1a1* mouse. b) Proliferation of GFP+ tumours in *Scgb1a1* (Club) and *Sftpc* (AT2) lineage tracing mice, measured by the percentage of BrdU+ cells per tumour analysed at endpoint. Club n=55 tumours analysed and AT2 n=137 tumours in at least 3 mice per condition . ****p<0.0001, unpaired t test.

Finally, we took advantage of a paper published last month in which AT2 derived *Kras* mutant;*p53* null tumours were sequenced using 10X scRNA-seq (Yang et al., Cell 2022). We decided to integrate our scRNAseq data with Yang's et al. data and exchanged the results in the manuscript. This new integration clearly shows that our Club-derived tumour cells cluster separate from their AT2 derived tumour cells (Revised Figures S7G and S7H). We also investigated how the gene expression module scores of our tumour clusters are distributed among our clusters and the ones from Yang et al. (Revised Figure 7E). We found that both the AT1-like tumour and the Regeneration-like tumour modules were significantly higher in our clusters and were most similar to the AT1 like and Late Gastric clusters from Yang et al.

We hope this reviewer agrees that this new integration looks much better than the one previously presented with the Marjanovic data and important conclusions can be taken, pointing that our Club tumour clusters do not overlap with the AT2 ones.

We really hope that all these arguments together will allow this reviewer to reconsider his/her decision and recommend this manuscript for publication.

Reviewer #3 (Remarks to the Author): expertise in mouse models of lung adenocarcinoma
Overall, the manuscript has improved and I am happy to recommend its acceptance.

We thank this reviewer and are pleased that he/she is happy to recommend acceptance of the manuscript.

REVIEWERS' COMMENTS

Reviewer #1 (Remarks to the Author):

My comments are all addressed.

Minor comments:

Figure legends for the supplementary figures 7G and H are switched. G is colored by Yang's cluster and H by current study.

June 30th 2022

Point-by-point rebuttal

REVIEWERS' COMMENTS

Reviewer #1 (Remarks to the Author):

My comments are all addressed.

Minor comments:

Figure legends for the supplementary figures 7G and H are switched. G is colored by Yang's cluster and H by current study.

We thank the reviewer for realizing this mistake. This has now been corrected.